# Distinct hierarchical alterations of intrinsic neural timescales account for different manifestations of psychosis

**Kenneth Wengler[1,2]\*, Andrew T Goldberg[2], George Chahine[3], Guillermo Horga[1,2]\***

[1]Department of Psychiatry, Columbia University, New York, United States; [2]New York State Psychiatric Institute, New York, United States; [3]Department of Psychiatry, Yale University, New Haven, United States

**Abstract** Hierarchical perceptual-inference models of psychosis may provide a holistic framework for understanding psychosis in schizophrenia including heterogeneity in clinical presentations. Particularly, hypothesized alterations at distinct levels of the perceptual-inference hierarchy may explain why hallucinations and delusions tend to cluster together yet sometimes manifest in isolation. To test this, we used a recently developed resting-state fMRI measure of intrinsic neural timescale (INT), which reflects the time window of neural integration and captures hierarchical brain gradients. In analyses examining extended sensory hierarchies that we first validated, we found distinct hierarchical INT alterations for hallucinations versus delusions in the auditory and somatosensory systems, thus providing support for hierarchical perceptual-inference models of psychosis. Simulations using a large-scale biophysical model suggested local elevations of excitation-inhibition ratio at different hierarchical levels as a potential mechanism. More generally, our work highlights the robustness and utility of INT for studying hierarchical processes relevant to basic and clinical neuroscience.

*For correspondence:
kenneth.wengler@nyspi.columbia.edu (KW);
HorgaG@nyspi.columbia.edu (GH)

**Competing interests:** The authors declare that no competing interests exist.

## Introduction

Hallucinations and delusions are burdensome symptoms that typically manifest together as the psychotic syndrome of schizophrenia. Perceptual-inference models of psychosis suggest that these symptoms result from alterations in the updating of internal models of the environment that are used to make inferences about external sensory events and their causes (*Adams et al., 2013*; *Horga and Abi-Dargham, 2019*; *Sterzer et al., 2018*). These models are receiving increasing empirical support (*Adams et al., 2018*; *Baker et al., 2019*; *Cassidy et al., 2018*; *Davies et al., 2018*; *Powers et al., 2017*; *Teufel et al., 2015*), yet current theories do not provide a satisfactory explanation for how hallucinations and delusions tend to co-occur but sometimes manifest in isolation. This suggests that these psychotic symptoms may share a common neurobiological mechanism while simultaneously depending on symptom-specific pathways.

We and others have proposed that this apparent tension may be resolved in the context of *hierarchical* perceptual-inference models (*Adams et al., 2013*; *Baker et al., 2019*; *Corlett et al., 2009*; *Corlett et al., 2019*; *Fletcher and Frith, 2009*; *Sterzer et al., 2018*). One possibility is that alterations at higher levels—supporting inferences on abstract hidden states like someone's intentions—may drive delusions, and alterations at lower levels—supporting inferences about lower level features of stimuli such as stimulus presence or absence—may drive hallucinations (*Baker et al., 2019*; *Davies et al., 2018*; *Horga and Abi-Dargham, 2019*; *Powers et al., 2017*). In addition to these symptom-specific pathways, alterations at any level may naturally propagate throughout the interdependent levels of the hierarchy (*Chaudhuri et al., 2015*), potentially explaining symptom co-

occurrence. Importantly, neural systems supporting inference are thought to feature a hierarchical architecture of timescales that mirrors the hierarchical temporal dynamics of natural environments, where rapidly changing events are typically nested within slower changing contexts (*Kiebel et al., 2008*; *Kiebel, 2009*). Thus, higher level inferences pertaining to slower changing contexts require neural systems with the ability to integrate information over longer periods, an ability consistent with the persistent neuronal activity that characterizes higher level regions (*Major and Tank, 2004*; *Mazurek et al., 2003*).

A hierarchy of neural timescales is observed in both single-neuron recordings in non-human primates (*Murray et al., 2014*) and functional magnetic resonance imaging (fMRI) recordings in humans (*Hasson et al., 2015*; *Hasson et al., 2008*; *Honey et al., 2012*; *Lerner et al., 2011*; *Stephens et al., 2013*), and is recapitulated by a large-scale biophysical model (*Chaudhuri et al., 2015*). Furthermore, a newly developed method based on resting-state fMRI that was validated against electroencephalography (EEG) similarly captures a hierarchy of intrinsic neural timescales (INT), as well as alterations in psychopathology (*Watanabe et al., 2019*). Here, we specifically applied this fMRI measure to test whether hallucinations and delusions are associated with distinct changes of INT at low and high levels of neural hierarchies, respectively. We hypothesized that INT at these respective levels would increase with more severe symptoms, potentially reflecting increased neural integration of prior information (*Glaze et al., 2015*; *Mante et al., 2013*; *Mazurek et al., 2003*), in line with behavioral findings in hallucinations and delusions (*Baker et al., 2019*; *Cassidy et al., 2018*; *Powers et al., 2017*). If present, these INT changes should manifest as symptom-specific differences in hierarchical gradients.

## Results

INT maps were estimated as previously described (*Watanabe et al., 2019*) (Materials and methods). Briefly, the autocorrelation function of the fMRI signal at each voxel (or vertex) was estimated and the sum of the autocorrelation coefficients during the initial positive period was calculated. This initial positive period included all timepoints from the first lag until the timepoint immediately preceding the first lagged timepoint with a non-positive autocorrelation coefficient. To adjust for differences in temporal resolution, the sum was multiplied by the repetition time (TR) of the fMRI data. This product was used as an index for INT (note that values are similar to those from an exponential fit [*Murray et al., 2014*; *Figure 1—figure supplement 1*]). INT maps were parcellated using the HCP-multimodal parcellation (*Glasser et al., 2016*) to facilitate further analysis. Additionally, T1w/T2w (myelin) and cortical-thickness maps were obtained from high-resolution structural scans from the HCP database. Both of these structural measures have previously been shown to capture an underlying brain-wide hierarchy (*Burt et al., 2018*; *Fischl et al., 2008*; *Wagstyl et al., 2015*), consistent with the classic use of myeloarchitecture and cytoarchitecture for cortical parcellation (*Brodmann, 1909*; *Sarkissov et al., 1955*; *Vogt, 1911*; *Von Economo, 1929*). In particular, *Burt et al., 2018* validated T1w/T2w in macaques by showing strong agreement with a gold-standard tract-tracing measure of hierarchy. Establishing T1w/T2w and cortical thickness as structural indices of hierarchy in humans, Burt et al. validated these MRI measures against human postmortem gene-expression data (*Hawrylycz et al., 2012*)—specifically using granular layer-IV-specific gene expression as a proxy for cytoarchitecture structural type.

### Selection and multimodal validation of neural hierarchies

Our hypothesis of symptom-specific INT differences in hierarchical gradients was agnostic to the specific neural hierarchies involved in psychosis, as the involvement of most sensory modalities has been reported (*Lewandowski et al., 2009*; *Postmes et al., 2014*). Consistent with prior empirical and theoretical work (*Chaudhuri et al., 2015*; *Vázquez-Rodríguez et al., 2019*), in a subset of 100 unrelated young and healthy subjects from the HCP dataset, we did not observe a systematic relationship across the whole brain between INT (*Figure 1A*)—an index of functional hierarchy (*Chaudhuri et al., 2015*; *Murray et al., 2014*)—and the two indices of structural hierarchy (T1w/T2w and cortical thickness) (*Burt et al., 2018*; *Fischl et al., 2008*; *Wagstyl et al., 2015*; *Figure 1—figure supplement 2*). We thus decided to focus on specific, well-studied hierarchies of the auditory, visual, and somatosensory systems that have been parcellated in humans, and where the functional and structural indices of hierarchy appeared better aligned.

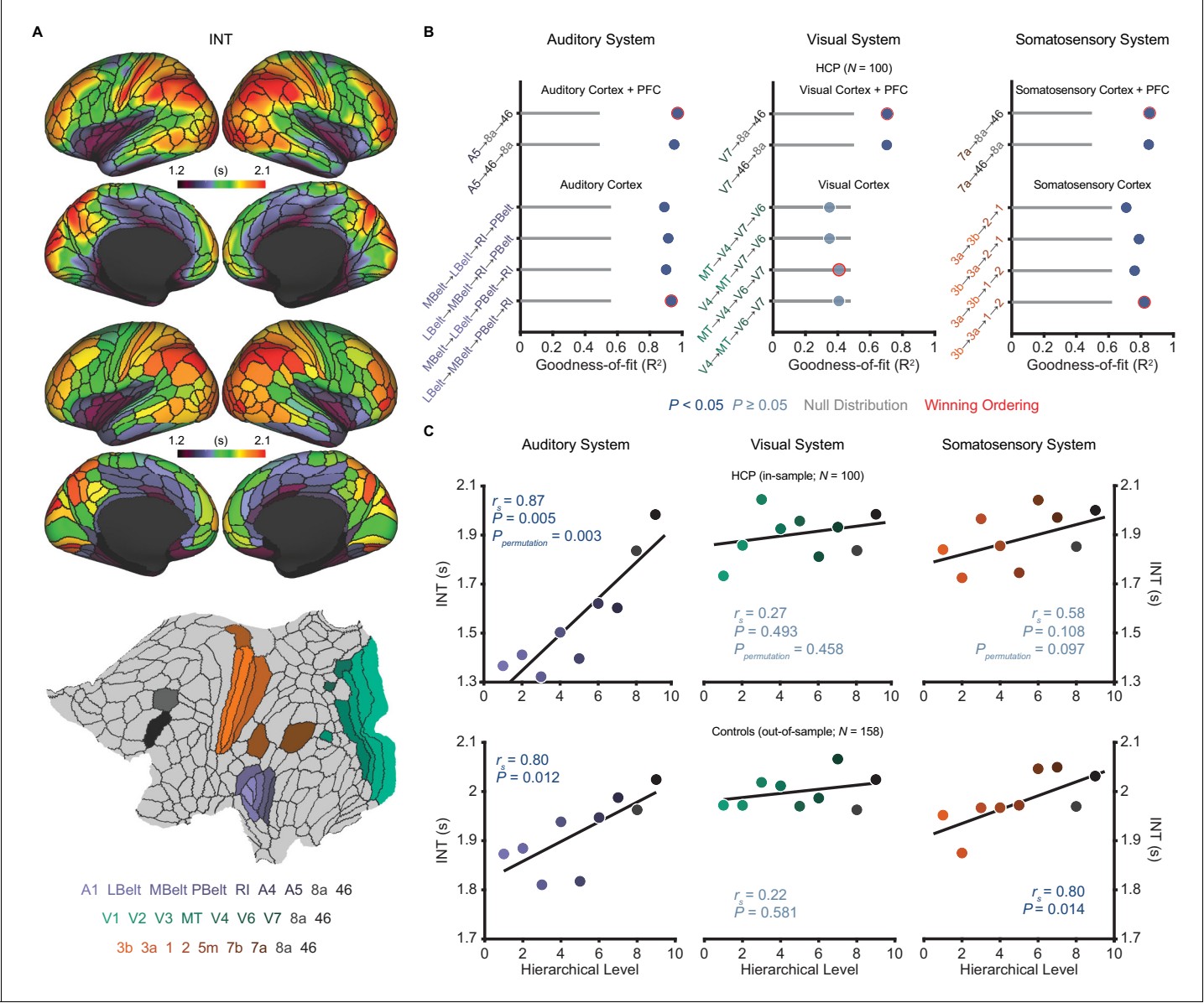

**Figure 1.** Model comparison to determine the hierarchical orderings of the auditory, visual, and somatosensory systems. (**A**) Group-averaged intrinsic neural timescale (INT) map from the Human Connectome Project (HCP) dataset ($N$ = 100; top), parcellated group-averaged INT map (middle), and flattened cortex showing the parcels in the auditory, visual, and somatosensory hierarchies (winning hierarchies underneath; bottom). Color coding of parcels indicates their anatomical and hierarchical location. (**B**) Goodness of fit ($R^2$) of linear mixed-effects models predicting different hierarchical orderings of the auditory system (left), visual system (middle), and somatosensory system (right) from T1w/T2w and cortical thickness values in the HCP dataset (Materials and methods). First, the winning ordering (i.e. the model with the best goodness of fit) for each system was determined for the seven sensory cortex regions (bottom four models). Then, the winning ordering was determined for extended models with two downstream prefrontal cortex regions added to the respective winning models for the sensory cortex (top two models). Note that, for each of the four considered orderings within the sensory cortex for each system, only the four regions whose order is varied (out of 7 regions) are shown to delineate the models. For the auditory cortex, A1 was always the lowest order region while A4 and A5 were always the two highest order regions. For the visual cortex, V1, V2, and V3 were always the three lowest order regions. For the somatosensory cortex, 5m, 7b, and 7a were always the three highest order regions. Null distributions were generated by randomly permuting the hierarchical ordering across all regions in a given hierarchy ($0^{th}$ – $95^{th}$ percentiles shown). (**C**) Scatterplots showing INT values plotted as a function of hierarchical level for the PFC-extended winning models in B (red outline) for the HCP dataset (top) and the healthy control group in the schizophrenia combined dataset ($N$ = 158; bottom). LBelt, lateral belt; MBelt, medial belt; PBelt, parabelt; RI, retroinsular cortex; MT, middle temproal area.

The online version of this article includes the following source data and figure supplement(s) for figure 1:

**Source data 1.** Data and code to reproduce *Figure 1*.

**Figure supplement 1.** Comparison of different methods for neural timescale estimation: sum of initial positive period vs exponential fit.

*Figure 1 continued on next page*

Despite ample anatomical investigation in primates, ambiguities in the definition of anatomical hierarchies in these sensory systems remain (*Hilgetag et al., 1996*; *Kaas and Hackett, 2000*) and have not been fully addressed in human MRI work. To address this issue, we used an anatomically informed, data-driven approach to determine the most suitable hierarchical orderings of each sensory system. First, using the HCP dataset, we determined the hierarchical orderings of the sensory cortex parcels (auditory, visual, or somatosensory) by selecting the ordering that was best predicted by T1w/T2w and cortical thickness parcel-wise values for each system (i.e. the ordering for which these values explained the most variance). To enhance robustness, we specifically constrained this comparison to the four most plausible hierarchical orderings of each system based on previous anatomical studies (*Felleman and Van Essen, 1991*; *Galaburda and Pandya, 1983*; *Hyvärinen and Poranen, 1978*; *Morel et al., 1993*). The winning orderings were A1 → lateral belt (LBelt) → medial belt (MBelt) → parabelt (PBelt) → retroinsular cortex (RI) → A4 → A5 for the auditory cortex, V1 → V2 → V3 → middle temporal area (MT) → V4 → V6 → V7 for the visual cortex, and 3b → 3a → 1 → 2 → 5m → 7b → 7a for the somatosensory cortex (*Figure 1B*). Using the same approach to build upon these winning orderings and capture the broadest possible range of the hierarchies, we then determined the hierarchical position of two additional prefrontal cortex (PFC) regions known to be downstream projections of the auditory, visual, and somatosensory cortices: area 8a and area 46 (*Felleman and Van Essen, 1991*; *Kaas and Hackett, 2000*). For all three sensory systems, area 46 was selected as the highest hierarchical level (*Figure 1B*). Notably, each of the PFC-extended winning models explained more variance than chance based on a null distribution of 10,000 random orderings (auditory system: $P_{permutation} < 10^{-4}$; visual system: $P_{permutation} = 0.003$; somatosensory system: $P_{permutation} = 0.001$).

We then evaluated whether these winning hierarchies—selected solely based on structural measures of hierarchy—were able to capture functional variability in the INT measure, such that higher levels exhibit longer INT. Within the HCP dataset, hierarchical position significantly correlated with INT in the auditory system (Spearman correlation $r_s = 0.87$, p=0.005; *Figure 1C*), and this correlation was above chance level based on a null distribution of 10,000 random orderings ($P_{permutation} = 0.003$). The hierarchical ordering was further validated in an out-of-sample group of 158 healthy controls from the schizophrenia combined dataset (Materials and methods), where hierarchy similarly correlated with INT in the auditory system ($r_s = 0.80$, p=0.01; *Figure 1C*). Positive but non-significant correlations were observed in the visual system (in-sample: $r_s = 0.27$, p=0.49, $P_{permutation} = 0.47$; out-of-sample: $r_s = 0.22$, p=0.58; *Figure 1C*). Stronger positive correlations were observed in the somatosensory system that reached significance in the out-of-sample group (in-sample: $r_s = 0.58$, p=0.108, $P_{permutation} = 0.097$; out-of-sample: $r_s = 0.80$, p=0.014; *Figure 1C*). Despite the non-significant effects in the visual system (which surprisingly seemed to reflect less pronounced hierarchical gradients on all MRI measures, as suggested by the structural MRI gradients for all four tested orderings of the visual cortex falling within the null distribution; *Figure 1B*), these data showed that the winning hierarchies captured functional INT gradients, at least in the auditory and somatosensory systems. As a third independent test of our winning hierarchies, we tested their ability to capture variability in cytoarchitecture structural type using human postmortem gene-expression data from the Allen Human Brain Atlas (*Hawrylycz et al., 2012*). Following prior work (*Burt et al., 2018*), we focused on the average expression of five genes preferentially expressed in granular layer IV, a cytoarchitectural marker that is more prominent in lower hierarchical levels. Consistent with this, expression of granular layer IV genes showed strong, negative correlations with hierarchical level in all three winning hierarchies (auditory: $r_s = -0.88$, p=0.003; visual: $r_s = -0.75$, p=0.026; somatosensory: $r_s = -0.87$, p=0.005; *Figure 1—figure supplement 3*). Thus, we empirically validated extended sensory hierarchies that captured variability in hierarchical indices across three independent datasets, although this was generally less clear for the visual system.

## Assessment of robustness and reliability in the HCP dataset

Next, we set out to determine the robustness and reliability of INT. We focused on head motion, a common source of artifacts in fMRI data (*Power et al., 2012*). Head motion during data acquisition was associated with decreased INT values (181 out of 188 parcels, $P_{permutation}$ = 0.01; *Figure 1—figure supplement 4*). Yet, these effects were comparable across hierarchical levels (auditory system: $r_s$ = −0.23, $P_{permutation}$ = 0.805; visual system: $r_s$ = −0.38, $P_{permutation}$ = 0.649; somatosensory system: $r_s$ = −0.88, $P_{permutation}$ = 0.054; *Figure 1—figure supplement 4*). No effects were observed for gender or age (all $P_{permutation}$ > 0.174). Finally, INT maps showed excellent reliability between the first and last 5 min of the fMRI acquisition (median ICC(2,1) ± interquartile range across voxels: 0.94 ± 0.03; *Figure 1—figure supplement 5*).

## Exploratory analyses of INT in schizophrenia versus health

Although our primary hypothesis dealt with hierarchical differences between hallucinations and delusions, we first present exploratory analyses of diagnosis effects on INT. *Table 1* lists the participant characteristics. Compared to controls (*N* = 158), patients (*N* = 127) exhibited a small-to-moderate, but widespread, reduction of INT (98 out of 188 parcels, $P_{permutation}$ = 0.013; *Figure 2E*). A voxelwise analysis observed a similar result (*Figure 2—figure supplement 1*). However, the INT reductions in patients were comparable across hierarchical levels (all $P_{permutation}$ > 0.40; *Figure 2F*). In silico simulations using a large-scale biophysical model (*Chaudhuri et al., 2015*) suggested that the global INT reduction in patients could be neuronally implemented by globally reduced excitation-inhibition (E/I) ratio (*Figure 2—figure supplement 2*).

## Hierarchical differences in INT between hallucinations and delusions

Our a priori hypothesis was that hallucinations and delusions are associated with alterations of INT at different hierarchical levels, leading to distinct changes in hierarchical gradients. To test this, we

**Table 1.** Participant characteristics.

| Variable | Healthy controls | | | | | Patients with schizophrenia | | | | |
|---|---|---|---|---|---|---|---|---|---|---|
| | BGS | COBRE | NMCH | UCLA | All | BGS | COBRE | NMCH | UCLA | All |
| N | 24 | 42 | 25 | 67 | 158 | 40 | 31 | 26 | 30 | 127 |
| Age, mean (SD), y | 36.0 (13.1) | 33.9 (10.4) | 29.8 (7.2) | 32.3 (8.5) | 32.9 (9.7) | 31.1 (12.6) | 30.5 (11.9) | 30.5 (6.2) | 35.0 (8.9) | 31.8 (10.6) |
| Male sex, No. (%) | 22 (92) | 34 (81) | 16 (64) | 54 (81) | 126 (80) | 38 (95) | 26 (84) | 19 (73) | 22 (73) | 105 (83) |
| Framewise displacement*, mean (SD), mm | 0.15 (0.04) | 0.14 (0.05) | 0.14 (0.11) | 0.10 (0.04) | 0.13 (0.06) | 0.16 (0.06) | 0.17 (0.06) | 0.12 (0.06) | 0.13 (0.04) | 0.15 (0.06) |
| Delusion score, mean (SD) | NA | NA | NA | NA | NA | 2.3 (1.6) | 1.7 (1.5) | 3.2 (1.9) | 2.5 (1.4) | 2.4 (1.7) |
| Hallucination score, mean (SD) | NA | NA | NA | NA | NA | 2.1 (1.6) | 1.7 (1.4) | 2.9 (2.0) | 2.2 (1.6) | 2.2 (1.7) |
| Conceptual disorganization score, mean (SD) | NA | NA | NA | NA | NA | 0.9 (1.3) | 0.6 (1.0) | 2.0 (1.6) | 1.4 (1.4) | 1.2 (1.4) |
| Emotional withdrawal score, mean (SD) | NA | NA | NA | NA | NA | 1.8 (1.2) | 1.2 (1.3) | 3.4 (1.7) | 2.3 (1.5) | 2.1 (1.6) |
| Social withdrawal score, mean (SD) | NA | NA | NA | NA | NA | 1.8 (1.4) | 1.3 (1.4) | 3.2 (1.7) | 2.7 (1.6) | 2.2 (1.6) |
| Blunted affect score, mean (SD) | NA | NA | NA | NA | NA | 1.8 (1.6) | 1.6 (1.5) | 3.3 (1.6) | 1.1 (1.1) | 1.9 (1.6) |
| Alogia score, mean (SD) | NA | NA | NA | NA | NA | 1.3 (1.6) | 1.2 (1.4) | 2.1 (1.7) | 0.9 (1.6) | 1.3 (1.6) |

*Framewise displacement values were estimated after motion censoring.

BGS, BrainGluSchi; NMCH, NMorphCH; SD, standard deviation.

The online version of this article includes the following source data for Table 1:

Source data 1. Raw data for each individual subject in *Table 1*.

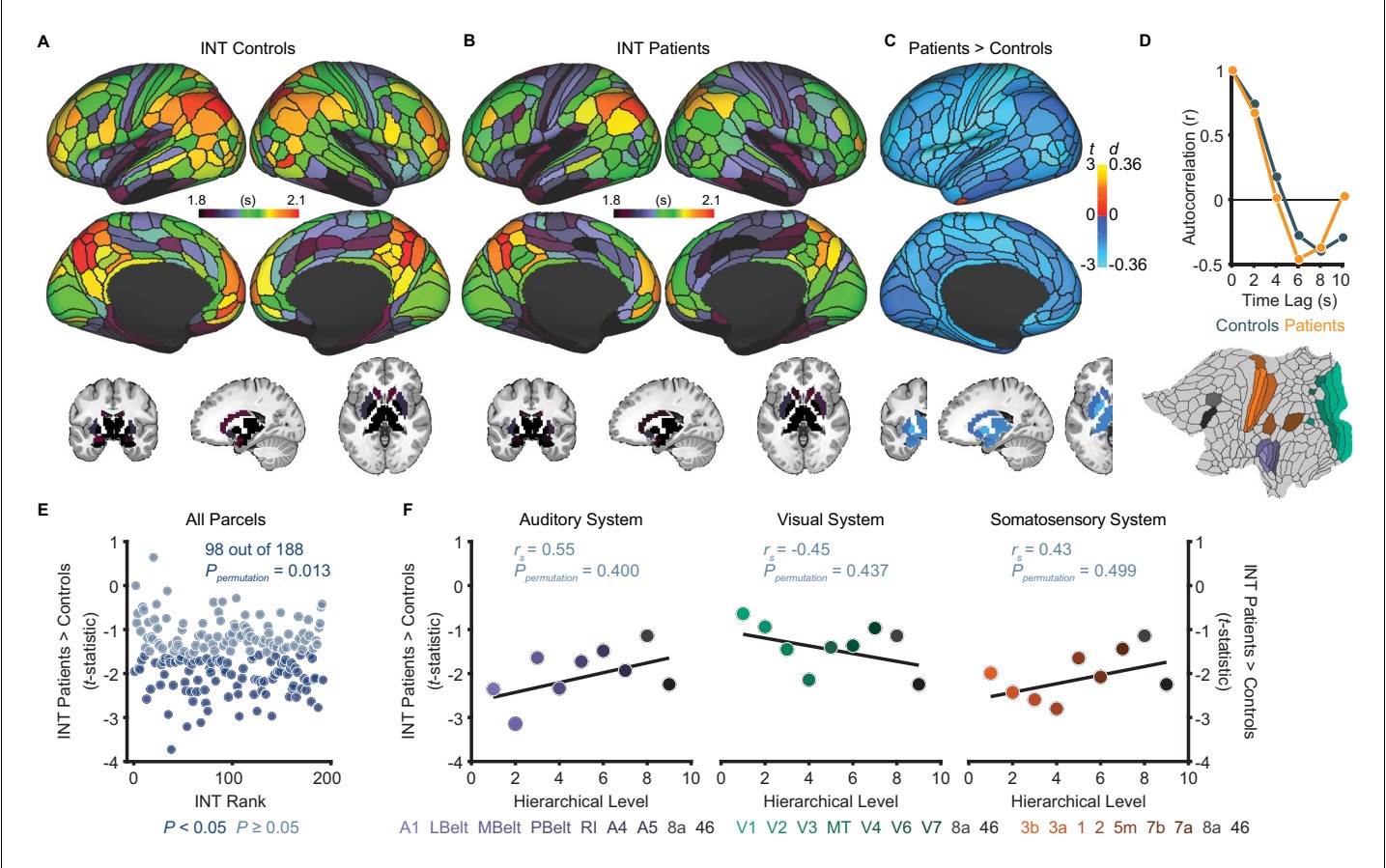

**Figure 2.** Exploratory analyses show that patients with schizophrenia exhibit widespread reductions of intrinsic neural timescales compared to healthy controls. (A) Parcellated group-averaged intrinsic neural timescale (INT) map for healthy controls ($N$ = 158). (B) Parcellated group-averaged INT map for patients with schizophrenia ($N$ = 127). (C) $t$-statistic (Cohen's $d$) map showing the contrast of patients greater than controls. Across most parcels, INT is shorter in patients than controls in a regression model (M1$_{exploratory}$; Materials and methods) controlling for age, gender, mean framewise displacement, and sample-acquisition site (overall effect of diagnosis: 98 out of 188 parcels, $P_{permutation}$ = 0.013). Only the left hemisphere is shown because statistical analyses were performed after averaging the values in each parcel across the left and right hemispheres. (D) To illustrate the effect of reduced INT in patients with schizophrenia, the group-averaged, whole-brain-averaged autocorrelation functions were estimated from subjects with fMRI data acquired with the same repetition time (top; controls: $N$ = 132; patients: $N$ = 101). The group-averaged autocorrelation function for patients crosses the zero point on the y-axis (i.e. autocorrelation coefficient = 0) sooner than in controls, demonstrating the global reduction of INT in patients. The flattened cortex shows the parcels in the auditory, visual, and somatosensory hierarchies for reference (bottom). (E) Scatterplot showing $t$-statistic values for group differences from the regression model (M1$_{exploratory}$), plotted as a function of the INT rank (determined from the group-averaged INT map from HCP subjects). Each datapoint represents one parcel. (F) Scatterplots showing $t$-statistic values from parcels within the auditory (left), visual (middle), and somatosensory (right) hierarchies plotted as a function of hierarchical level. No hierarchical-gradient effects of schizophrenia diagnosis were observed. LBelt, lateral belt; MBelt, medial belt; PBelt, parabelt; RI, retroinsular cortex; MT, middle temproal area.

The online version of this article includes the following source data and figure supplement(s) for figure 2:

**Source data 1.** Data and code to reproduce *Figure 2*.
**Figure supplement 1.** Voxelwise analysis of intrinsic neural timescales in schizophrenia versus health.
**Figure supplement 2.** Biophysical model simulation of reduced intrinsic neural timescales in schizophrenia versus health.

determined the unique variance associated with the effect of interindividual variability in hallucination and delusion severity on INT (M1$_{primary}$; Materials and methods). As expected, severity of hallucinations and delusions in our sample were correlated ($r_s$ = 0.62, p<0.01) but had sufficient unique variance [$(1 - R^2)$=0.62] to evaluate their independent contributions. The severity of these symptoms was uncorrelated with antipsychotic dose among the 109 patients with available data (chlorpromazine equivalents: both p>0.86), making medication an unlikely confound (*Figure 3—figure supplement 1*).

The model (M2; Materials and methods) that we used as a primary test of main effects and interactions of symptoms on hierarchical INT gradients—which also included interaction terms for each sensory system to account for between-system differences—was significant (omnibus $F_{11,41}$ = 5.52, p<$10^{-4}$). Critically, within this model we found hierarchical-gradient effects that differed significantly between hallucinations and delusions in the expected directions for 2/3 systems (auditory system, symptom-by-hierarchical-level interaction: $t_{42}$ = 4.59 [95% bootstrap confidence interval: 3.39, 9.08], Cohen's $f^2$ = 1.00, $P_{permutation}$ = 0.001; visual: $t_{42}$ = −2.06 [−6.19, 0.16], $f^2$ = 0.11, $P_{permutation}$ = 0.083; and somatosensory: $t_{42}$ = 3.50 [2.19, 7.35], $f^2$ = 0.41, $P_{permutation}$ = 0.011; *Figure 3*). In the auditory system, this interaction was driven by significant hierarchical-gradient effects in opposite directions for hallucinations (hierarchical-level effect: $t_{42}$ = −3.50 [−8.42,−2.24], $f^2$ = 0.41, $P_{permutation}$ = 0.010) and delusions (hierarchical-level effect: $t_{42}$ = 2.99 [1.18, 6.37], $f^2$ = 0.27, $P_{permutation}$ = 0.025). In the somatosensory system, this effect was driven by a trend-level negative hierarchical-gradient effect for hallucinations (hierarchical-level effect: $t_{42}$ = −2.35 [−5.91,−1.00], $f^2$ = 0.15, $P_{permutation}$ = 0.056) and a significant positive hierarchical-gradient effect for delusions (hierarchical-level effect: $t_{42}$ = 2.60 [1.43, 5.57], $f^2$ = 0.19, $P_{permutation}$ = 0.042; *Figure 3*). In the visual system, hierarchical-gradient effects were not significant for either symptom (hallucination hierarchical-level effect: $t_{42}$ = 0.90 [−1.06, 3.62], $f^2$ = 0.00, $P_{permutation}$ = 0.466; delusion hierarchical-level effect: $t_{42}$ = −2.01 [−6.01, 0.53], $f^2$ = 0.11, $P_{permutation}$ = 0.087). We also found significant interactions with sensory system, indicating differences in the symptom interactions between the visual and the other systems (see statistics in *Figure 3B*), but these were not a priori tests (see also Discussion for issues of interpretability). Examining the significant symptom effects further, in the auditory system we observed that patients with high-severity hallucinations exhibited a numeric increase in INT at lower levels of the hierarchy relative to those with low-severity hallucinations, leading to a compression of the INT hierarchical gradient (*Figure 3C*); in contrast, in both the auditory and somatosensory systems, patients with high-severity delusions exhibited a numeric increase in INT at higher levels of the hierarchy relative to those with low-severity delusions, leading to a more pronounced INT hierarchical gradient.

To correct for multiple comparisons, we carried out a family-wise permutation test determining the probability of spuriously obtaining the set of significant effects we observed in support of our hypothesis. Based on the chance level of jointly observing negative hierarchical-gradient effects of hallucination severity in at least 1/3 systems, *and* positive hierarchical-gradient effects of delusion severity in at least 2/3 systems, *and* interaction effects of hierarchy-by-symptom in the expected direction in at least 2/3 systems, this analysis suggested that the observed set of results was statistically above chance (set-level $P_{permutation}$ = 0.014). Furthermore, based on the chance level of observing a significant negative hierarchical-gradient effect for hallucinations, *and* a significant positive hierarchical-gradient effect for delusions, *and* a significant symptom-by-hierarchical-level interaction (i.e. all three effects in one system), the observed set of results in the auditory system was also statistically above chance (set-level $P_{permutation}$ = 0.043).

To rule out an effect of our approach for selecting hierarchical orderings on these results, we tested these symptom effects for each of the four different sensory cortex hierarchical orderings considered a priori candidates for each sensory system. Results were generally consistent across the different hierarchical orderings (*Figure 3—figure supplement 2*), particularly in the auditory system. A family-wise permutation test similar to the one above, but including all four orderings per system (12 total orderings), showed that the observed set of results was statistically above chance for all systems (set-level $P_{permutation}$ = 0.002) and for the auditory system alone (set-level $P_{permutation}$ = 0.001).

## Post-hoc analysis of the specificity of INT hierarchical-gradient effects

In a post-hoc analysis, we then investigated the specificity of these hierarchical-gradient effects to the positive psychotic symptoms under investigation. To this end, we determined hierarchical-gradient effects individually for each symptom in the auditory and somatosensory systems using a model including symptom-severity effect (only one symptom), hierarchical level, sensory system (auditory, visual, and somatosensory), and their interactions. In the auditory system, conceptual disorganization was the only symptom—other than hallucinations and delusions—that showed a significant effect (hierarchical-level effect: $t_{21}$ = −2.80 [−5.31,−1.10], $f^2$ = 0.60, $P_{permutation}$ = 0.036; *Figure 3—figure supplement 3*). But this effect was weaker than that for hallucinations (hierarchical-level effect: $t_{21}$ = −4.38 [−10.31,−3.28], $f^2$ = 10.57, $P_{permutation}$ = 0.005; *Figure 3—figure supplement 3*). These

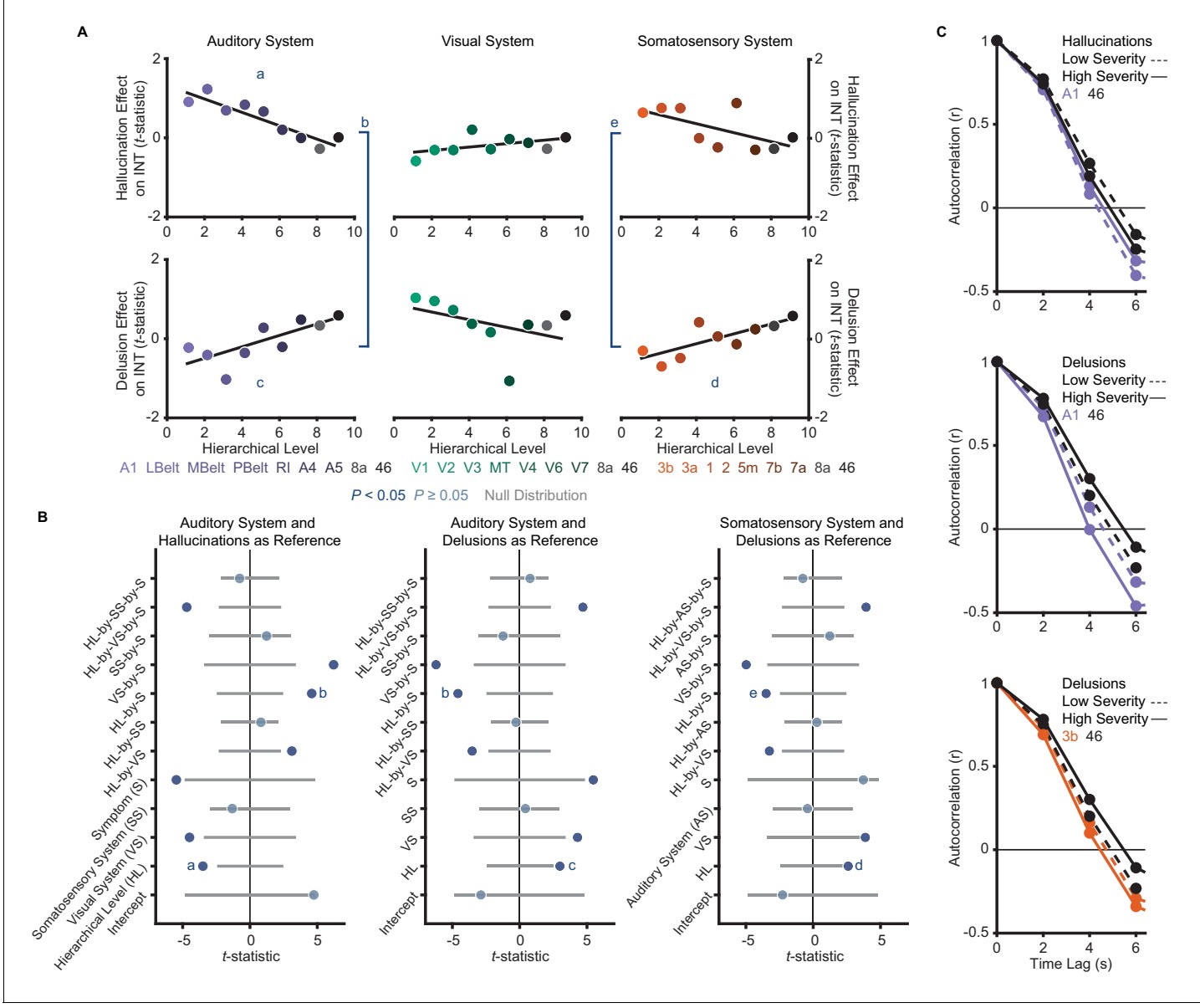

**Figure 3.** A priori analyses show that hallucinations and delusions exhibit distinct hierarchical-gradient effects on intrinsic neural timescales in the auditory and somatosensory systems. (**A**) Scatterplots showing *t*-statistic values from a regression model (M1$_{primary}$; Materials and methods) including all seven symptoms and controlling for age, gender, mean framewise displacement, and sample-acquisition site for hallucination-severity (top) and delusion-severity (bottom) effects from parcels within the auditory (left), visual (middle), or somatosensory (right) systems plotted as a function of hierarchical level (using PFC-extended winning hierarchies; *Figure 1*). (**B**) Summary of results from a model (M2; Materials and methods) including symptom-severity effect (hallucinations or delusions), hierarchical level, sensory system (auditory, visual, or somatosensory), and their interactions. These results demonstrate: (1) in the auditory system, a significant difference in the relationship between hallucination severity and hierarchical level versus that for delusion severity and hierarchical level (b); (2) in the auditory system, significant hierarchical-gradient effects of hallucination severity (a) and delusion severity (c); (3) in the somatosensory system, a significant difference in the relationship between hallucination severity and hierarchical level versus that for delusion severity and hierarchical level (e); and (4) in the somatosensory system, a significant hierarchical-gradient effect of delusion severity (d). Note that different symptoms and systems were used as references (implicit variable) across three plots to show each of the relevant effects which were tested within a single model (M2). Null distributions were generated by randomly permuting symptom-severity scores across patients in M1$_{primary}$ (2.5$^{th}$ – 97.5$^{th}$ percentiles shown). (**C**) To illustrate the effects, the group-averaged autocorrelation functions were estimated from subjects with fMRI data acquired with the same repetition time (*N* = 10 for each group). High-severity patients were the 10 subjects with the highest residual symptom scores after regressing out all other symptoms; low-severity patients were the 10 subjects with the lowest residual symptom scores. The group-averaged autocorrelation functions are shown for high-severity (solid lines) and low-severity (dashed lines) hallucination patients from low and high levels of the auditory hierarchy (A1 and area 46, top). The group-averaged autocorrelation functions are also shown for high-severity and low-severity delusion patients from low and high levels of the auditory hierarchy (middle). Finally, the group-averaged autocorrelation functions are shown

*Figure 3 continued on next page*

*Figure 3 continued*

for high-severity and low-severity delusion patients from low and high levels of the somatosensory hierarchy (area 3b and area 46, bottom). These plots depict a compression of the auditory hierarchical gradient in high-severity hallucination patients and, instead, an expansion of both the auditory and somatosensory hierarchical gradients in high-severity delusion patients. LBelt, lateral belt; MBelt, medial belt; PBelt, parabelt; RI, retroinsular cortex; MT, middle temproal area.

The online version of this article includes the following source data and figure supplement(s) for figure 3:

**Source data 1.** Data and code to reproduce *Figure 3*.
**Figure supplement 1.** Controlling for antipsychotic dose does not change the distinct hierarchical-gradient effects of hallucination and delusion severity in the auditory and somatosensory systems.
**Figure supplement 2.** Distinct hierarchical-gradient effects of hallucinations and delusions are robust to the choice of sensory hierarchies.
**Figure supplement 3.** Only positive symptoms show hierarchical-gradient effects in the auditory system and only delusions show a hierarchical-gradient effect in the somatosensory system.
**Figure supplement 4.** Comparison of hierarchical-gradient effects in the dorsal and ventral auditory systems.
**Figure supplement 5.** Distinct hierarchical-gradient effects of hallucination and delusion severity in the auditory system are observed when using an anatomically agnostic definition of the hierarchy.

results thus suggest some specificity to positive symptoms, which conceptual disorganization is classically defined as (*Association AP, 2013*; *VandenBos, 2007*) (but see *van der Gaag et al., 2006*), consistent with a stronger correlation of conceptual disorganization with positive symptoms (average $r_s$ = 0.48) versus negative symptoms (average $r_s$ = 0.23) in our sample. Indeed, a permutation test comparing the average strength of hierarchical-gradient effects (i.e. mean absolute-value of $t$-statistics) for positive versus negative symptoms (i.e. blunted affect, social withdrawal, emotional withdrawal, and alogia) showed the effects of positive symptoms to be significantly larger than the effects of negative symptoms ($P_{permutation}$ = 0.043) in the auditory system. In the somatosensory system, no symptoms other than delusions showed a significant hierarchical-gradient effect. Hallucinations, however, showed the strongest negative effect (hierarchical-level effect: $t_{21}$ = −2.23 [−6.79,−1.75], $f^2$ = 0.31, $P_{permutation}$ = 0.079; *Figure 3—figure supplement 3*).

Thus, although the hierarchical-gradient effects were not unique to the two symptoms under investigation—which is not required under perceptual-inference models of psychosis and which could suggest model extensions to account for additional phenomena—these effects were strongest for, and relatively specific to, positive symptoms.

## Altered E/I ratio as a potential biological mechanism

To explore candidate biological mechanisms for the effects we observed in vivo, we leveraged a large-scale biophysical model previously shown to capture intrinsic timescale hierarchies (*Chaudhuri et al., 2015*). This model depicts the macaque cortex using 29 recurrently connected nodes, with connection strengths based on macaque tract-tracing studies (*Figure 4B*). Given growing evidence for E/I imbalance in schizophrenia (*Foss-Feig et al., 2017*; *Jardri et al., 2016*) and the hypothesized local increases of INT, we fit the biophysical model to our data to explore whether our results could be driven by local increases in E/I ratio. These E/I ratio changes were modeled as a triangle function where a local maximum exhibited a peak E/I ratio increase and other nodes had E/I ratio changes that decreased linearly as a function of absolute distance in hierarchical levels from the peak. This function was described by three free parameters: (i) the hierarchical level of the peak E/I ratio increase, (ii) the magnitude of the E/I ratio increase at the peak, and (iii) the magnitude of the E/I ratio change at the minimum (i.e. at the hierarchical level furthest from the peak).

To fit the biophysical model, we first estimated in vivo data for 'exemplary cases' using regression fits from M1$_{primary}$ (Materials and methods) in the auditory system—the system that showed the strongest effects in vivo. The regression fits allowed us to estimate INT values at each level of the hierarchy for exemplary cases representative of extreme symptom profiles (while controlling for variability in other factors). INT values for the auditory hierarchy were estimated for four exemplary cases: (1) no hallucinations or delusions (fitted INT values from M1$_{primary}$ with minimum scores of 0 for both symptoms); (2) hallucinations only (maximum score of 5 for hallucinations and score of 0 for delusions); (3) delusions only (scores of 0 for hallucinations and 5 for delusions); (4) hallucinations and delusions (scores of 5 for both symptoms). For all exemplary cases, the severity of other symptoms and the values of covariates were set to the average values from all patients. Changes of INT for

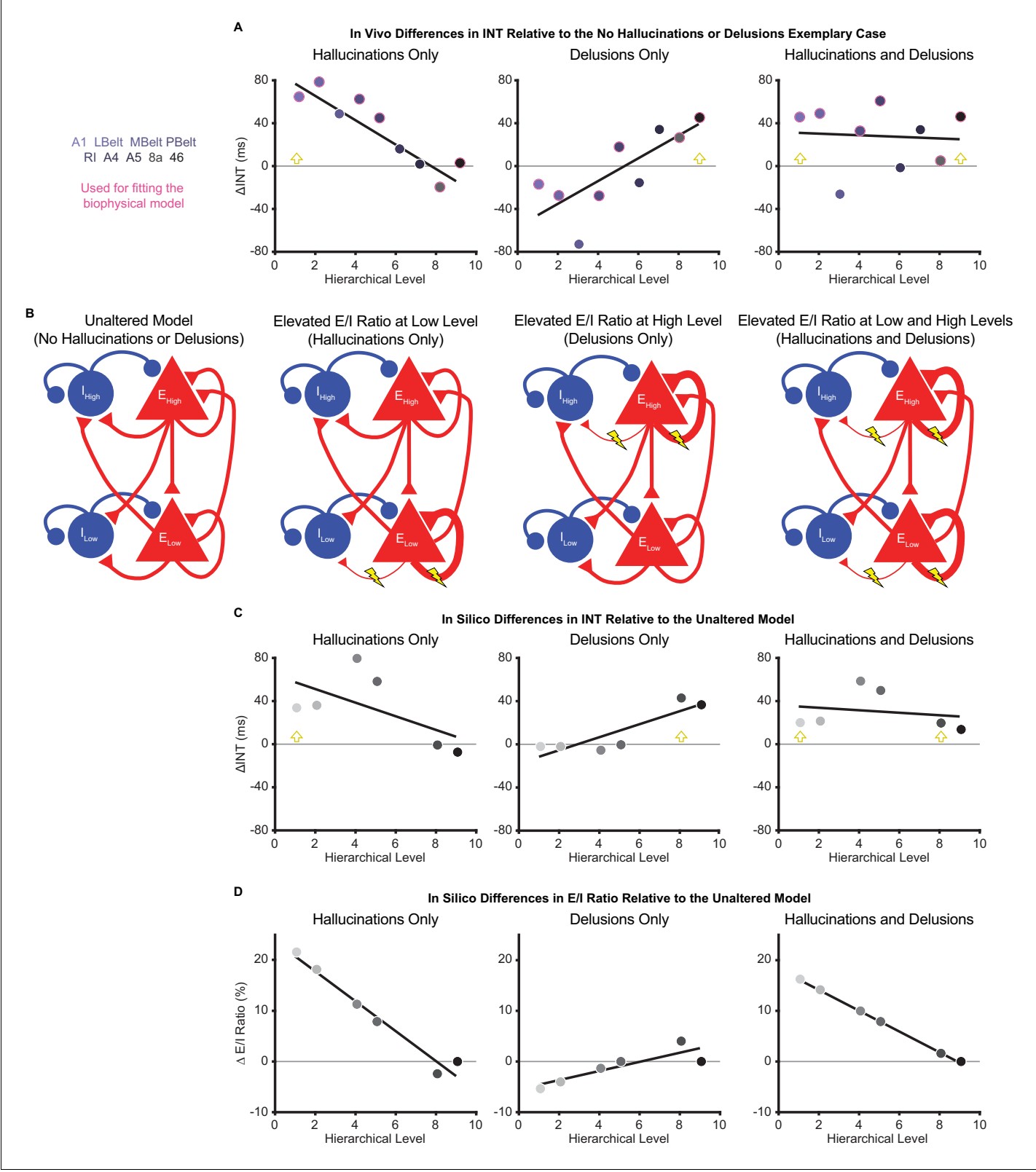

**Figure 4.** Hallucination and delusion effects on intrinsic neural timescales are recapitulated by elevated excitation-inhibition ratios at different hierarchical levels. (A) Scatterplots showing the difference in estimated intrinsic nerual timescale (INT) values between the three exemplary cases capturing extreme symptom profiles ('hallucinations only', 'delusions only', and 'hallucinations and delusions') with respect to the 'no hallucinations or delusions' exemplary case (in vivo ΔINT; fitted parcel-wise data from M1$_{primary}$). The parcel data used for fitting the biophysical model are outlined in

*Figure 4 continued on next page*

*Figure 4 continued*

pink. Yellow arrowheads denote the hypothesized hierarchical level of the maximum excitation-inhibition (E/I) ratio increase. (**B**) Simplified schematic of a large-scale biophysical model of the macaque cortex and its variants (Materials and methods). The model consists of 29 nodes with local excitatory (red triangles) and inhibitory (blue circles) pools of neurons; only two nodes—high (top) and low (bottom) hierarchical levels—are shown for illustrative purposes. These nodes have both local (recurrent) and long-range (across-node) connections. Lightning bolts mark theoretical perturbations to the model. Thicker or thinner lines with respect to the reference scenario reflect increased or decreased connection strengths, respectively. Note that E/I ratio can be increased either by increasing local excitatory-to-excitatory connection strength *or* by decreasing local excitatory-to-inhibitory connection strength, but these scenarios were modeled individually. (**C**) Scatterplots showing the difference in simulated INT values between each of the three pathological biophysical models ('elevated E/I ratio at low level', 'elevated E/I ratio at high level', and 'elevated E/I ratio at low and high levels') with respect to the reference biophysical model ('unaltered model') using the best-fitting E/I ratio parameters (in silico ΔINT). By allowing the E/I ratios to vary, the biophysical model can recapitulate the in vivo INT changes with a negative (compressed) hierarchical-gradient effect for hallucinations, a positive (expanded) hierarchical-gradient effect for delusions, and an overall INT increase (without a manifest hierarchical-gradient effect) for the combined case of hallucinations and delusions. The visual hierarchy (V1, V2, V4, MT, 8l, and 46d) was used for these simulations given the lack of tract-tracing data for the auditory cortex—other levels of the visual cortex are omitted for the same reason—but the qualitative pattern of results applies to hierarchical-gradient effects in any given sensory system. Yellow arrowheads denote the hierarchical level of the maximum E/I ratio increase. (**D**) Scatterplots showing the fitted changes to E/I ratios for the pathological biophysical models: the in vivo INT changes associated with hallucinations can be recapitulated by elevated E/I ratio at the lowest hierarchical level and those associated with delusions by elevated E/I ratio (of smaller magnitude) at the highest hierarchical level, with the addition of these two alterations capturing the changes in patients with both hallucinations and delusions. Note that E/I ratio in level 9 of the hierarchy was fixed to its value in the unaltered model to prevent model instability (Materials and methods). LBelt, lateral belt; MBelt, medial belt; PBelt, parabelt; RI, retroinsular cortex.

The online version of this article includes the following source data for figure 4:

**Source data 1.** Data and code to reproduce *Figure 4*.

exemplary cases 2–4 were determined as the difference in INT relative to the 'no hallucinations or delusions' case (in vivo ΔINT; *Figure 4A*). We modeled the in vivo ΔINT in the auditory system using the macaque visual system as a model hierarchy with realistic biological constraints due to the lack of tract-tracing data for the auditory system; note that sensory system and species differences limit our ability to derive precise quantitative conclusions from the modeling results but still afford qualitative insights. We specifically used the six biophysical-model nodes that directly corresponded to levels of our visual hierarchy and for which tract-tracing data were available: V1 (level 1), V2 (level 2), V4 (level 4), MT (level 5), 8l (level 8), and 46d (level 9). Model-derived in silico ΔINT (*Figure 4C*) were calculated for each node as the difference in INT from the biophysical model with no perturbations ('unaltered model') and the INT from the best-fitting biophysical model for which the values of the parameters controlling the E/I ratio provided the closest approximation to the in vivo ΔINT across exemplary cases (in the six corresponding parcels of our auditory hierarchy: A1 [level 1], LBelt [level 2], PBelt [level 4], RI [level 5], 8a [level 8], and 46 [level 9]). Specifically, two sets of the 3 E/I ratio parameters were jointly fitted to exemplary cases 2–4, one for hallucinations and one for delusions, with the combined effect of hallucinations and delusions resulting from the sum of the E/I ratio changes for the two individual symptoms.

In silico results using the best-fitting parameters were able to recapitulate the INT effects of hallucinations and delusions (compression versus expansion of the INT hierarchical gradient, respectively) via local increases in E/I ratio at low or high levels of the hierarchy, respectively. Specifically, the best-fitting levels of the peak increase in local E/I ratio were levels 1 and 8 for hallucinations and delusions, respectively (*Figure 4D*). Interestingly, given the relatively greater strength of both recurrent and long-range connections at higher levels that is built into the biophysical model, the required peak E/I ratio increase to achieve the observed changes of INT was considerably smaller for the delusion-related alteration at level 8 (ΔE/I = 4.02%) compared to the hallucination-related alteration at level 1 (ΔE/I = 21.61%). Also, the in silico ΔINT based on the summed E/I ratio alterations for individual symptoms closely approximated the combined case of hallucinations and delusions (exemplary case 4), which consisted of a general increase in INT with no clear change in the hierarchical gradient. This suggests that additivity of the local symptom-specific alterations could explain symptom co-occurrence.

In a follow-up analysis, we further explored our in vivo data for evidence of the additive effect of hallucinations and delusions, focusing on the auditory system. We first compared the average INT across all auditory system parcels between patients with both high hallucination and delusion scores (i.e. raw average data from subjects with a score of 5 for both symptoms; *N* = 11) and patients with

neither hallucinations nor delusions (i.e. subjects with a score of 0 for both symptoms; $N$ = 18). Here, we observed significantly higher average INT in patients with high-severity hallucinations and delusions ($t_{27}$ = 1.84, p=0.038; one-tailed two-sample t-test). Second, we fit a linear model predicting auditory parcel INT as a function of hierarchical level, allowing separate intercepts for each of the two groups, and an interaction between hierarchical level and group. Here, we found that the intercept was indeed higher for patients with high-severity hallucinations and delusions compared to patients with neither symptom ($t_{257}$ = 2.04, p=0.043). Furthermore, we found no difference in the hierarchical gradients between these groups, with a non-significant hierarchical-level-by-group interaction ($t_{257}$ = 0.65, p=0.519). Although preliminary, these results supply some support for the notion of additive hierarchical alterations in psychosis.

## Control analyses examining alternative definitions of auditory hierarchies

Given that the hierarchical-gradient effects supporting our initial hypotheses were clearest in the auditory system—a system thought to comprise dual processing streams—we considered the impact of alternative definitions of the auditory hierarchy on our results.

Diverging auditory streams with downstream projections to dorsal (areas 8a and 46) versus ventral (areas 10 and 12vl) PFC have been described (*Kaas and Hackett, 2000*). We thus conducted an ordering-selection analysis for the ventral stream, like that presented above for the dorsal stream (Selection and Multimodal Validation of Neural Hierarchies). Within the ventral stream, area 12vl was better predicted as the highest hierarchical level, and the winning ordering explained more variance than chance based on a null distribution of random orderings (ventral auditory system: $P_{permutation}$ < $10^{-4}$; *Figure 1—figure supplement 6A*). Furthermore, similar to the dorsal stream, hierarchical level in the ventral stream correlated with INT (in-sample: $r_s$ = 0.87, p=0.005, $P_{permutation}$ = 0.003; out-of-sample: $r_s$ = 0.80, p=0.014; *Figure 1—figure supplement 6B*) and expression of granular layer IV genes ($r_s$ = −0.91, p=0.001). Given these dual auditory streams and their corresponding validated hierarchies, we explored potential differences in hierarchical-gradient effects of hallucination and delusion severity for the dorsal versus ventral streams. The model explaining symptom effects and their differences by hierarchical level and their interaction by symptoms and auditory stream was significant (omnibus $F_{7,27}$ = 8.60, p<$10^{-4}$). We further found a significant difference in the symptom-by-hierarchical-level effects between the dorsal and ventral auditory streams (symptom-by-hierarchical-level-by-processing-stream interaction: $t_{28}$ = 1.75 [1.02, 3.90], $f^2$ = 0.12, $P_{permutation}$ = 0.005; *Figure 3—figure supplement 4*). In the ventral stream, we did not find a significant difference in the hierarchical-gradient effects between hallucinations and delusions (symptom-by-hierarchical-level interaction: $t_{28}$ = 2.01 [−0.37, 6.43], $f^2$ = 0.17, $P_{permutation}$ = 0.159; *Figure 3—figure supplement 4*); we found a trend-level hierarchical-gradient effect of hallucination severity (hierarchical-level effect: $t_{28}$ = −2.56 [−7.04,−0.73], $f^2$ = 0.31, $P_{permutation}$ = 0.098; *Figure 3—figure supplement 4*) and no effect of delusion severity (hierarchical-level effect: $t_{28}$ = 0.28 [−2.42, 3.36], $f^2$ = 0.00, $P_{permutation}$ = 0.445; *Figure 3—figure supplement 4*). Interestingly, comparing the dorsal and ventral auditory streams, we observed a significant difference in the hierarchical-gradient effect of delusion severity (hierarchical-level-by-processing-stream interaction: $t_{28}$ = 1.89 [1.39,3.49], $f^2$ = 0.15, $P_{permutation}$ = 0.003; *Figure 3—figure supplement 4*). These results thus support the involvement of dorsolateral PFC in delusions, consistent with prior work (*Corlett et al., 2007*).

As an additional control for the uncertainty in defining the auditory hierarchy, we also adopted an anatomically agnostic, data-driven approach. First, the symptom effects (M1$_{primary}$) were estimated for each voxel within the nine auditory-system parcels (600 voxels total). Second, each voxel was ranked based on its INT value in the average INT map from the HCP dataset. Third, 10 equally spaced bins along the INT ranking (60 voxels per bin) were created, which comprised the levels of the data-driven hierarchy, and the voxelwise $t$-statistics (from M1$_{primary}$) were averaged per bin. Similar to the main analysis (*Figure 3*), a model that included main effects and interactions of symptoms on the hierarchical INT gradient was significant (omnibus $F_{4,15}$ = 10.20, p=0.001). Within this model, we found hierarchical-gradient effects that differed significantly between hallucinations and delusions (symptom-by-hierarchical-level interaction: $t_{16}$ = 5.19 [0.48 23.26], $f^2$ = 10.12, $P_{permutation}$ = 0.003; *Figure 3—figure supplement 5*). This interaction was driven by significant hierarchical-gradient effects in opposite directions for hallucinations (hierarchical-level effect: $t_{16}$ = −2.79 [−16.72, 2.40], $f^2$ = 0.25, $P_{permutation}$ = 0.049; *Figure 3—figure supplement 5*) and delusions

(hierarchical-level effect: $t_{16}$ = 4.55, [1.04, 19.76], $f^2$ = 4.04, $P_{permutation}$ = 0.008; *Figure 3—figure supplement 5*).

## Discussion

Using a recently developed method for measuring neural timescales from resting-state fMRI data, we set out to test the hypothesis that hallucinations and delusions are associated with dysfunctions at different levels of neural hierarchies. Using established structural indices of hierarchy (myelin and cortical thickness) and INT (a functional index of hierarchy) in independent samples, we first validated extended sensory hierarchies for the auditory, visual, and somatosensory systems that captured substantial variability in the hierarchical MRI indices. After further showing excellent reliability of the INT measure, in exploratory analyses, we showed for the first time that patients with schizophrenia have globally reduced INT. Most importantly, our primary analyses comparing INT effects for hallucinations versus delusions in the validated hierarchies demonstrated that these symptoms are associated with distinct changes of the hierarchical gradients in the auditory and somatosensory systems, an effect we failed to observe in the visual system.

Hierarchical models of perceptual inference posit that perceptions are shaped by prior beliefs (*Dayan et al., 1995*; *Friston and Kiebel, 2009*; *Kiebel, 2009*; *Lee and Mumford, 2003*; *Rao and Ballard, 1999*) through reciprocal message passing across different levels of sensory hierarchies, an architecture that mirrors the known anatomy of sensory systems (*Felleman and Van Essen, 1991*; *Glasser et al., 2016*; *Kaas and Hackett, 2000*; *Markov et al., 2014a*; *Van Essen et al., 1992*; *Young, 1993*). In this scheme, higher levels of the neural hierarchy are thought to represent increasingly abstract belief states that evolve at slower timescales (*Kiebel, 2009*). For instance, during speech perception, the hierarchical structure of linguistic units can be parsed such that lower levels of auditory processing encode syllable information at faster timescales while higher levels encode sentence information at slower timescales (*Ding et al., 2016*). An emerging body of work in psychosis has linked hallucinations to preferential biases toward prior beliefs in low-level inferences during detection or estimation of stimulus features (*Cassidy et al., 2018*; *Davies et al., 2018*; *Powers et al., 2017*; *Teufel et al., 2015*) and delusions to preferential biases toward prior beliefs in higher-level inferences about more abstract, hidden states (*Baker et al., 2019*). The observed biases toward prior beliefs in past behavioral work can be framed as primacy biases (*Baker et al., 2019*), where past information is weighted more heavily during the inferential process, or equivalently, where information is integrated over longer timescales (*Glaze et al., 2015*). Temporal integration is at the core of the neural implementation of perceptual inference (*Mazurek et al., 2003*) and is thought to depend crucially on recurrent network activity (*Chaudhuri et al., 2015*; *Mante et al., 2013*). Thus, a plausible neuronal implementation of primacy biases at a given level of the hierarchy would be through increases in the strength of recurrent excitation or decreases in the strength of recurrent inhibition (i.e. elevated E/I ratio) leading to relative increases in neural timescales.

Here, we observed changes in neural timescales across levels of neural hierarchies that differed between hallucinations and delusions, an effect that was most evident in the auditory hierarchy. Patients with more severe hallucinations exhibited a less pronounced INT hierarchical gradient, consistent with increased timescales at lower levels compared to those with less severe hallucinations; those with more severe delusions instead exhibited a more pronounced INT hierarchical gradient, consistent with increased timescales at higher levels compared to those with less severe delusions (*Figure 3C*). We further recapitulated these findings by respectively elevating E/I ratios at low or high hierarchical levels of a large-scale biophysical model (*Chaudhuri et al., 2015*). These E/I ratio elevations could, in principle, result from alterations in NMDA or dopamine activation at these levels and are thus plausible under widely supported glutamatergic and dopaminergic theories of psychosis (*Brunel and Wang, 2001*; *Corlett et al., 2009*; *Corlett et al., 2011*; *Durstewitz and Seamans, 2002*; *Jardri et al., 2016*; *Javitt et al., 2012*; *Weinstein et al., 2017*). These results thus demonstrate distinct hierarchical alterations for hallucinations and delusions that are generally consistent with our hypothesized hierarchical framework, where distinct hierarchical alterations provide symptom-specific pathways that together may explain symptom co-occurrence, thus providing a candidate biological mechanism for the psychotic syndrome.

Such hierarchical alterations may also fit well with the phenomenological timescale of these symptoms. Clinical observation indicates that hallucinations—like rapidly changing sensory events—

change transiently and intermittently over seconds or minutes, while delusions—like slowly changing 'conceptual' beliefs—evolve more slowly over days or months, but their average severities over a given period typically evolve in parallel. These clinical features are consistent with a hierarchical structure of nested timescales (*Kiebel et al., 2008*). While our findings generally support this notion, computational work explicitly laying out the proposed model in the context of inferential alterations in psychosis and empirical confirmations are warranted. One outstanding question is how the delusion-related alterations in neural timescales we observed—which may predominate in high levels of the hierarchy but manifest as changes on the order of seconds—might drive delusions evolving over much longer timescales. One possible explanation is that, while delusion maintenance may involve long-term memory processes, the underlying mechanism initiating delusions transpires more rapidly and disrupts inferences at timescales on the order of seconds, consistent with prior work (*Baker et al., 2019*). Since encoded memories likely reflect inferences summarizing information at a given timepoint (*Shadlen and Shohamy, 2016*), high-level inferential biases at shorter timescales may be sufficient to shape long-term conceptual memories in a way that further propagates biases over long time-periods, particularly under primacy biases that decrease the relative influence of newer information. Although less critical, it is also worth noting that INT reflects differences in resting circuit dynamics, the timescale of which is likely to be substantially magnified when these circuits are engaged (*Chaudhuri et al., 2015*; *Hasson et al., 2008*).

Our opposing findings for diagnosis (globally reduced INT) and symptom severity (focally increased INT) may be reconciled within pathophysiological models of psychosis which posit a key role for compensatory processes in schizophrenia. Hallucinations and delusions have been proposed to represent a temporary state of the illness that results from a failed attempt to compensate for a trait-like, baseline deficit (*Adams et al., 2013*; *Moutoussis et al., 2011*). Relatedly, long-standing circuit-level theories have suggested that psychosis-related increases in striatal dopamine transmission are secondary to a primary cortical deficit (*Weinberger, 1987*). In particular, previous frameworks suggest that psychotic states are associated with excessive prior biases in inferential processes arising as an overcompensation for a baseline trait consisting of the opposite bias (*Adams et al., 2013*; *Horga and Abi-Dargham, 2019*). From a biophysical-modeling standpoint, the trait-like baseline deficit in schizophrenia could consist of globally reduced E/I ratio (for instance, arising from NMDA-receptor hypofunction of excitatory neurons *Cavanagh et al., 2019*), which behaviorally would translate into general recency biases. In contrast, a failed compensatory mechanism could result in local increases in E/I ratio at different levels leading to distinct primacy biases and psychotic symptoms (*Lam et al., 2017*). While speculative, the compensatory changes could arise from dopaminergic alterations that effectively increase E/I ratio by preferentially boosting NMDA-receptor function of excitatory neurons (or other changes dampening NMDA-receptor function of inhibitory neurons) (*Brunel and Wang, 2001*).

Our finding of preferential involvement of the auditory system for hallucinations is not surprising, given that in schizophrenia this symptom tends to predominate in the auditory modality despite also presenting in other modalities (*Lim et al., 2016*; *Waters and Fernyhough, 2017*); auditory-cortex abnormalities in schizophrenia are also well established (*Javitt and Sweet, 2015*). Our finding of somatosensory system involvement for delusions is also consistent with previous work on delusions of passivity (*Brüne et al., 2008*; *Spence et al., 1997*) and deficits in sensory attenuation via motor predictions in schizophrenia (*Shergill et al., 2005*; *Shergill et al., 2014*). However, despite our failure to detect differential alterations in the visual system, substantial evidence also suggests visual-cortex abnormalities in schizophrenia (*Butler et al., 2008*; *Cavuş et al., 2012*; *Dorph-Petersen et al., 2007*). And evidence from subclinical populations suggests symptom-specific hierarchical alterations in visual tasks (*Davies et al., 2018*). Furthermore, the general differences in INT values between sensory systems (*Figure 1*), while potentially relevant to psychosis in and of themselves, could imply differential sensitivity in our analyses across sensory domains. Our null findings in the visual system are also qualified by the poorer correspondence between levels of the visual hierarchy and hierarchical MRI indices (not only for INT but also surprisingly for the structural indices) compared to the other systems (*Figure 1*). This suggests the need for further investigation into the sensitivity of available MRI measures of hierarchy to uncover the underlying gradients within the visual cortex.

Previous empirical work using structural (*Bassett et al., 2008*) and functional measures (*Dondé et al., 2019*; *Leitman et al., 2010*; *Yang et al., 2016*), suggests hierarchical alterations in

schizophrenia. This work, however, did not evaluate hierarchical differences between symptoms and used measures that differ fundamentally from INT. In exploratory analyses testing diagnostic effects, we found global INT reductions in schizophrenia but no clear shifts in the hierarchical INT gradients (see *Figure 2—figure supplement 2* for initial evidence of an exponential effect). We used the same approach as a previous study measuring INT in individuals with autism, which reported decreased INT in the visual cortex (and increased INT in the caudate) (*Watanabe et al., 2019*). Consistent with our interpretation, this INT phenotype was linked to other data in autism supporting excessive weighting of sensory evidence (*Gollo, 2019*; *Lawson et al., 2017*)—akin to a decreased primacy bias (i.e. a recency bias).

Some limitations are worth discussing. Because 93% of the patients (with available medication data) were taking antipsychotics, we cannot definitively rule out medication confounds, particularly on diagnosis effects. However, we observed similar effects when controlling for dose in our main analysis, no correlations between dose and symptoms, and did not expect differential neural effects on hallucinations versus delusions (*Figure 3—figure supplement 1*); future studies should elucidate medication effects on INT. Additionally, our study was limited to investigating the effects of global severity of hallucinations and delusions and could not resolve effects of symptom subtype or content, since detailed assessments were only available in a small subset of our patients. Larger studies with more detailed assessments are needed to tease out these potential effects.

In conclusion, we have presented evidence for distinct hierarchical alterations in neural timescales as a function of hallucination and delusion severity, lending initial neural support for hierarchical views of psychosis. Additionally, our work suggests that INT (*Watanabe et al., 2019*) provides a reliable and interpretable measure of neural function with the potential to elucidate hierarchical alterations and dysfunctions in circuit dynamics in schizophrenia and other neuropsychiatric disorders.

## Materials and methods

### Human Connectome Project dataset

T1w/T2w and cortical thickenss maps, and resting-state fMRI data were obtained for a subset of 100 unrelated young and healthy subjects from the Human Connectome Project (HCP) WU-Minn Consortium (*Van Essen et al., 2013*). The first fMRI run (single-shot EPI with left-to-right phase encoding direction) from the first fMRI session was obtained for each subject during eyes-open-on-fixation with the following scanning parameters: repetition time (TR) = 720 ms; spatial resolution = 2 × 2×2 mm; timepoints = 1200. High-resolution (0.7 mm isotropic voxels) T1w and T2w anatomical images were also acquired. Details regarding subject recruitment and MRI data acquisition have been previously reported (*Smith et al., 2013*; *Van Essen et al., 2012*). Preprocessing of the HCP data was performed using the HCP minimal preprocessing pipeline (*Glasser et al., 2013*). The preprocessed fMRI data were then used for the estimation of INT maps in 32k Conte69 mesh surface space and MNI152_ICBM2009a_nlin volume space with native spatial resolution. The T1w/T2w (myelin) maps (*Glasser and Van Essen, 2011*) in 32k Conte69 mesh surface space were used to compare functional (INT) and structural (T1w/T2w) measures of hierarchy.

### Schizophrenia combined dataset

T1w images and resting-state fMRI data were obtained for 331 healthy control subjects and 254 patients diagnosed with either schizophrenia (N = 241) or schizoaffective disorder (N = 13) from four publicly available datasets. Three of these datasets were from the SchizConnect repository [BrainGluSchi (*Bustillo et al., 2016*), COBRE (*Aine et al., 2017*; *Çetin et al., 2014*), and NMorphCH (*Alpert et al., 2016*)] and one was from the OpenfMRI repository (UCLA; *Poldrack et al., 2016*). Data that survived a quality-control check (~95%) and motion-censoring check (~64%) included 140 patients and 225 controls. The quality control check consisted of visual inspection of the spatially normalized images. The motion-censoring check consisted of determining if there were sufficient degrees of freedom after motion censoring to perform nuisance variable regression. A subset of 158 controls was then selected that matched patients on gender and age. To minimize scanner- and site-related differences, we excluded subjects if the signal-to-noise ratio (SNR) was less than 100 for any of the standard regions-of-interest (*Power et al., 2011*). The final sample after quality-control checks consisted of 127 patients and 152 age- and gender-matched controls (*Table 1*).

The fMRI data were collected for each subject during eyes-open-on-fixation with the following scanning parameters: TR = 2000 ms (except for NMorphCH, where TR = 2200 ms); timepoints (Brain-GluSchi/COBRE/NMorphCH/UCLA) = 165/150/323/152; spatial resolution (mm) = 3.5×3.5×3.5/3.5×3.5×3.5/4×4×4/3×3×3. Data were preprocessed using the AFNI afni_proc.py function (*Cox, 1996*). The following steps were performed: (1) removal of the first five volumes with the 3dTcat function; (2) slice-timing correction; (3) motion correction; (4) 12-parameter affine regis-tration of the fMRI images to the T1w image; (5) spatial normalization of fMRI images to MNI152_ICBM2009a_nlin volume space using nonlinear warping via the T1w image; (6) single-inter-polation resampling of fMRI images combining motion correction and spatial normalization.

Symptom severity in patients was assessed with the Positive and Negative Syndrome Scale (PANSS) (*Kay et al., 1987*) in the COBRE and BrainGluSchi samples, and with the Scale for the Assessment of Positive Symptoms (SAPS) (*Andreasen, 1984*) and the Scale for the Assessment of Negative Symptoms (SANS) (*Andreasen, 1983*) in the UCLA and NMorphCH samples. To appropri-ately combine the scores across all four samples, we chose the subset of seven items that consti-tuted unequivocal matches between the PANSS and SAPS/SANS (in parentheses): delusions (global rating of delusions), conceptual disorganization (global rating of positive formal thought disorder), hallucinatory behavior (global rating of hallucinations), blunted affect (global rating of affective flat-tening), emotional withdrawal (global rating of anhedonia/asociality), passive/apathetic social with-drawal (global rating of avolition/apathy), lack of spontaneity and flow of conversation (global rating of alogia). PANSS scores were decreased by one point for all levels of severity and the severe and moderately severe levels were combined into a single level so that scoring conformed to the SAPS/SANS scale (from 0 to 5 with increasing severity).

## Intrinsic neural timescale maps

Before estimating the voxelwise or vertexwise INT values, preprocessed fMRI data were further proc-essed with the following steps: (1) regression of white-matter signal, cerebrospinal-fluid signal, global-brain signal, and the six motion parameters along with their first derivatives; (2) bandpass fil-tering in the 0.01–0.1 Hz range; (3) motion censoring of volumes with framewise displacement (FD) (*Power et al., 2012*) greater than 0.3 mm along with the volumes directly preced-ing and following that volume; (4) spatial smoothing with a 4 mm full-width-at-half-maximum Gauss-ian kernel. INT maps were estimated following the procedure of *Watanabe et al., 2019*. At a single-participant level, the processed resting-state fMRI data were used to estimate the INT value in each voxel or vertex (for HCP dataset only). First, the autocorrelation function was estimated according to:

$$ACF_k = \frac{\sum_{t=k+1}^{T} (y_t - \bar{y})(y_{t-k} - \bar{y})}{\sum_{t=1}^{T} (y_t - \bar{y})^2} \tag{1}$$

where $k$ is the time lag, $T$ is the total number of timepoints, and $y$ is the fMRI signal. INT was then estimated as the area under the curve of the ACF during the intial positive period:

$$INT = TR \cdot \sum_{k=1}^{N} ACF_k \tag{2}$$

where $TR$ is the repetition time of the fMRI signal and $N$ is the lag directly preceeding the first nega-tive ACF value.

After INT estimation, the INT maps for subjects from the BrainGluSchi, COBRE, and NMorphCH samples were resampled to a spatial resolution of 3×3×3 mm to match the UCLA sample.

## HCP dataset analyses

Based on previous work showing that lower T1w/T2w map values co-localize with higher hierarchical levels (*Burt et al., 2018*), as do longer neural timescales (*Chaudhuri et al., 2015*; *Murray et al., 2014*), we examined the spatial relationship between T1w/T2w, cortical thickness, and INT values. We restricted this examination to the HCP dataset since its high-resolution and high-quality

structural MRI data allows for precise estimation of myelin maps. Group-averaged T1w/T2w, cortical thickness, and INT maps in surface space were parcellated using the HCP-multimodal parcellation (HCP-MMP1.0) (*Glasser et al., 2016*). The parcels were separated into either the six parcel groups that the 22 sections described by Glasser et al. are divided into: (1) visual (sections 1–5); (2) sensori-motor (sections 6–9); (3) auditory (sections 10–12); (4) remaining temporal cortex (sections 13–14); (5) remaining posterior cortex (sections 15–18); (6) remaining anterior cortex (sections 19–22) (*Glasser et al., 2016*); or 12 networks (*Ji et al., 2019*). We tested the parcel-wise spatial relationship between T1w/T2w or cortical thickness and INT values using linear regression.

Findings from the parcel-wise analysis did not support a brain-wide, system- or network-independent alignment of structural and functional hierarchies. This motivated a search for anatomically informed hierarchies within the sensory systems (auditory, visual, and somatosensory). Linear mixed-effects models were used to determine the best-fitting hierarchical ordering for each system. Models predicted hierarchical level from fixed- and random-effects (per subject) of T1w/T2w and cortical thickness values for parcels ordered accordingly. Hierarchical orderings were first determined for the sensory cortices. The four most likely orderings for each sensory system were determined based on the primate anatomy literature (*Felleman and Van Essen, 1991*; *Galaburda and Pandya, 1983*; *Hyvärinen and Poranen, 1978*; *Kaas and Hackett, 2000*; *Morel et al., 1993*). The auditory cortex regions were A1, lateral belt (LBelt), medial belt (MBelt), parabelt (PBelt), retroinsular cortex (RI), A4, and A5. The positions of LBelt and MBelt were allowed to take either level 2 or 3 of the hierarchy; PBelt and RI were allowed to take either level 4 or 5; A1 was level 1, A4 was level 6, and A5 was level 7 in all cases. The visual regions were V1, V2, V3, V4, middle temporal area (MT), V6, and V7. The positions of V4 and MT were allowed to take either level 4 or 5 of the hierarchy; V6 and V7 were allowed to take either level 6 or 7; V1 was level 1, V2 was level 2, and V3 was level 3 in all cases. The somatosensory cortex regions were areas 3b, 3a, 1, 2, 5m, 7b, and 7a. The positions of areas 3b and 3a were allowed to take either level 1 or 2 of the hierarchy; areas 1 and 2 were allowed to take either level 3 or 4; area 5m was level 5, area 7b was level 6, and area 7a was level 7 in all cases. Since all compared models had the same number of variables, the winning models for each system were simply determined based on the orderings that explained the most variance ($R^2$). After selection of the hierarchies in the sensory cortices, two downstream prefrontal cortex regions (areas 8a and 46) were added as either level 8 or 9 of the hierarchy based on a second model comparison. The PFC-extended winning hierarchies were then validated by determining the relationship of hierarchical levels with INT values using non-parametric Spearman correlations ($r_s$) both in the HCP (in-sample) dataset and in the control group from the schizophrenia combined (out-of-sample) dataset. Following prior work (*Burt et al., 2018*), the winning hierarchies were additionally validated against human postmortem gene-expression data from the Allen Human Brain Atlas (*Hawrylycz et al., 2012*).

## HCP robustness analyses

Because the schizophrenia combined dataset was analyzed in volume space, HCP dataset single-subject and group-averaged INT maps were also estimated in volume space and parcellated into 180 cortical parcels using HCP-MMP1.0 in volume space (https://identifiers.org/neurovault.collection: 1549) and 8 FreeSurfer (*Fischl, 2012*) subcortical parcels (thalamus, caudate, putamen, pallidum, hippocampus, amygdala, nucleus accumbens, and ventral diencephalon). The reliability of INT maps was assessed at the voxel level in volume space using the two-way random, single score intraclass correlation coefficient [ICC(2,1)] (*Shrout and Fleiss, 1979*). INT maps for each of the 100 subjects estimated using the first 5 min of data acquisition (similar to the amount of data available for the schizophrenia datasets) were compared to those estimated using the last 5 min of data acquisition of a single 14-min run. To evaluate potential confounds of the INT values, we examined their relationship with age, gender, and head motion—based on mean framewise displacement (*Power et al., 2012*) (FD)—using linear regression.

## Schizophrenia combined dataset analyses

INT maps were parcellated into 180 cortical parcels using HCP-MMP1.0 in volumetric space (https://identifiers.org/neurovault.collection:1549) and 8 FreeSurfer (*Fischl, 2012*) subcortical parcels (thalamus, caudate, putamen, pallidum, hippocampus, amygdala, nucleus accumbens, and ventral diencephalon). In an exploratory analysis, differences in INT map values between patients with

schizophrenia and healthy controls were investigated using a linear-regression model ($M1_{exploratory}$) predicting parcel-wise INT as a function of diagnosis while controlling for age, gender, mean FD, and sample-acquisition site (BrainGluSchi, COBRE, NMorphCH, and UCLA). To test our hypothesis of hallucination- and delusion-specific alterations of INT, we evaluated the relationships between symptom severity and INT values using a linear-regression model ($M1_{primary}$) predicting parcel-wise INT with each of the seven symptoms (hallucinations, delusions, conceptual disorganization, emotional withdrawal, social withdrawal, blunted affect, and alogia) as regressors while controlling for age, gender, mean FD, and sample-acquisition site. We did not use voxelwise statistical parametric mapping approaches because our main focus was on effects along hierarchical gradients not necessarily dependent on anatomical proximity. Our main test focused on differences between hallucinations and delusions in INT gradient effects within anatomically informed hierarchies of the auditory, visual, and somatosensory systems—reflecting symptom-specific INT alterations at different hierarchical levels. We specifically tested our primary hypothesis using a linear-regression model (M2) predicting auditory, visual, and somataosensroy system $t$-statistics for hallucination and delusion severity from $M1_{primary}$ as a function of symptom, hierarchical level, and sensory system. The interactions of symptom-by-hierarchical-level were used to directly test our hypothesis. This model included full interactions for all varaibles (symptoms, heirarchical level, and sensory systems). We included sensory-system interactions to allow for differences between sensory systems. A post-hoc power analysis for M2 showed our analyses had between 88% and 99% power to detect effect sizes (Cohen's $f^2$) between 0.19 and 0.36 ($\alpha$ = 0.05).

## Permutation testing

To assess statistical significance while controlling for multiple comparisons, we used permutation tests, which provide adequate protection against false positives in fMRI analyses (*Eklund et al., 2016*). Permutation tests compared observed effects ($t$-statistics of individual regression coefficients from M2 [or $M1_{exploratory}$]) to those in a null distribution obtained from 10,000 surrogate datasets in which the values of the predictor variables of interest in $M1_{primary}$ (or $M1_{exploratory}$) were randomly shuffled. Corrected p-values at 0.05 ('$P_{permutation}$'), two-sided, are reported. Permutation tests were also used to determine null distributions of the hierarchy model-comparison for determining the hierarchical orderings. There, null distributions were obtained from 10,000 surrogate datasets in which the hierarchical level of each region was randomly assigned. Corrected p-values at 0.05, one-sided, are reported for the model-comparison step while corrected p-values at 0.05, two-sided, are reported for the in-sample INT correlation.

## Bootstrap confidence intervals

Bootstrap confidence intervals were determined for the results from M2 using the accelerated bias-corrected ($BC_a$) percentile method (*Efron, 1987*). 10,000 bootstraps were performed at the level of $M1_{primary}$ and two-sided 95% confidence intervals were determined.

## Large-scale biophysical model of cortical neural timescales

We implemented the model of *Chaudhuri et al., 2015*, a large-scale biophysical model of hierarchical dynamic processing in the primate cortex. We chose this model because it was constructed using gold-standard tract-tracing experiments to determine the directed- and weighted-connectivity strengths between nodes (unlike similar models of the human cortex). Additionally, this model captures the observed hierarchy of intrinsic neural timescales *Murray et al., 2014*. The model contains 29 nodes, each consisting of an excitatory and inhibitory population. The populations are described by:

$$\tau_E \tfrac{d}{dt} v_E = -v_E + \beta_E [I_E]_+$$
$$\tau_I \tfrac{d}{dt} v_I = -v_I + \beta_I [I_I]_+$$

(3)

$v_E$ is the firing rate of the excitatory population, with intrinsic time constant $\tau_E$ and input current $I_E$, and for which the f-I curve has the slope $\beta_E$. $[I_E]_+ = \max(I_E, 0)$. The inhibitory population has corresponding parameters $v_I$, $\tau_I$, $I_I$ and $\beta_I$. Values for $\tau_E$, $\tau_I$, $\beta_E$, and $\beta_I$ are given below and taken from prior work (*Binzegger et al., 2009*).

At each node, the input currents have a component originating within the area (i.e. local input) and another originating from other areas (i.e. long-range input):

$$I_E^i = (1 + \eta h_i)\left(w_{EE}v_E^i + I_{lr,E}^i\right) - w_{EI}v_I^i + I_{ext,E}^i$$

$$I_I^i = (1 + \eta h_i)\left(w_{IE}v_E^i + I_{lr,I}^i\right) - w_{II}v_I^i + I_{ext,I}^i \tag{4}$$

The super- and sub-script, $i$, denotes the node (1 – 29), $w_{EE}$ and $w_{EI}$ are couplings to the excitatory population from the local excitatory and inhibitory population respectively, $I_{lr,E}^i$ is the long-range input to the excitatory population, and $I_{ext,E}^i$ is external input (both stimulus input and any noise added to the system). $w_{IE}$, $w_{II}$, $I_{lr,I}^i$, and $I_{ext,E}^i$ are the corresponding parameters for the inhibitory population.

The excitatory inputs to an area, both local and long-range, are scaled by its position in the hierarchy, $h_i$ (see below for details). $h_i$ is normalized between 0 and 1, and $\eta$ is a scaling parameter that controls the effect of hierarchy. By setting $\eta = 0$, the intrinsic differences between areas are removed. Note that both local and long-range projections were scaled by hierarchy, rather than just local projections, following prior observations (*Markov et al., 2011*).

Long-range input is modeled as excitatory current to both excitatory and inhibitory cells:

$$I_{lr,E}^i = \mu_{EE}\sum_{j=1}^{29} FLN_{ij}v_E^j$$

$$I_{lr,I}^i = \mu_{IE}\sum_{j=1}^{29} FLN_{ij}v_E^j \tag{5}$$

Here, $j$ ranges over all areas. $I_{lr,E}^i$ and $I_{lr,I}^i$ are the long-range inputs to the excitatory and inhibitory populations, $v_E^j$ is the firing rate of the excitatory population in area $j$ and $FLN_{ij}$ is the fraction of labeled neurons (FLN; see below for details) projecting from area $j$ to area $i$. $\mu_{EE}$ and $\mu_{IE}$ are scaling parameters that control the strengths of long-range input to the excitatory and inhibitory populations, respectively, and do not vary between connections; all the specificity comes from the FLN. Long-range connectivity is thus determined by three parameters: $\mu_{EE}$ and $\mu_{IE}$ control the connection strengths of long-range projections, and $\eta$ maps the hierarchy into excitatory connection strengths. The excitatory-to-inhibitory ratio of input current, $\gamma = I_{inp,E}/I_{inp,I}$, was chosen such that the steady-state firing rate of the excitatory population does not change when the current is present. Given an input of $I_{inp,E}$ to the excitatory population, an input of $\gamma I_{inp,E}$ to the inhibitory population increases the inhibitory firing rate sufficiently to cancel out the additional input to the excitatory population. $\mu_{EE}$ and $\mu_{IE}$ were chosen with a ratio slightly above this value so that projections are weakly excitatory.

Parameter values were: $\tau_E$ = 20 ms, $\tau_I$ = 10 ms, $\beta_E$ = 0.066 Hz/pA, $\beta_I$ = 0.351 Hz/pA, $w_{EE}$ = 24.3 pA/Hz, $w_{IE}$ = 12.2 pA/Hz, $w_{EI}$ = 19.7 pA/Hz, $w_{II}$ = 12.5 pA/Hz, $\mu_{EE}$ = 33.7 pA/Hz, $\mu_{IE}$ = 25.3 pA/Hz and $\eta$ = 0.68. Background input for each area was chosen so that the excitatory and inhibitory populations had rates of 10 and 35 Hz, respectively. As in Chaudhuri et al., we added an external input of white-noise to all areas with a mean of 0 Hz and a standard deviation of $10^{-5}$ Hz to simulate the resting-state condition.

Connectivity data are from an ongoing project that is quantitatively measuring all connections between cortical areas in the macaque cortex (*Markov et al., 2013*; *Markov et al., 2014a*). The connection strengths between areas are measured by counting the number of neurons labeled by retrograde tracer injections. To control for injection size, these counts are normalized by the total number of neurons labeled in the injection, giving a fraction of labeled neurons (FLN):

$$FLN_{j \to i} = \frac{\text{number of neurons projecting to area i from area j}}{\text{total number of neurons projecting to area i from all areas}} \tag{6}$$

These data were also used to estimate the fraction of neurons in a projection originating in the supragranular layers (SLN):

$$SLN_{j \to i} = \frac{\text{number of supragranular neurons projecting to area i from area j}}{\text{number of neurons projecting to area i from area j}} \tag{7}$$

The hierarchy was constructed following a similar framework to *Markov et al., 2014b*, using a generalized linear model. Hierarchical values were assigned to each area such that the difference in values predicts SLN (*Barone et al., 2000*):

$$SLN_{j \to i} \approx g^{-1}\left(h_i - h_j\right) \tag{8}$$

where $g^{-1}$ is a logistic function (logistic regression) and $h_i$ is the hierarchy value of area $i$. In the fit, the contribution of each projection is weighted by the log of its FLN to preferentially match stronger and less noisy projections (*Chaudhuri et al., 2015*). All connectivity data can be downloaded from www.core-nets.org.

The simulated neuronal activity was converted to blood-oxygen-level-dependent (BOLD) fMRI signal using the Balloon-Windkessel hemodynamic model (*Stephan et al., 2007*), a dynamical model that describes the transduction of neuronal activity ($v_E$) to changes in a vasodilatory signal ($s$) that is subject to autoregulatory feedback. This vasodilatory signal is coupled to changes in cerebral blood flow ($f$) that result in changes to the normalized total deoxyhemoglobin content ($q$) and normalized venous blood volume (v). For each area ($i$), these biophysical variables are defined by the following equations:

$$\frac{ds_i}{dt} = v_E^i - \kappa s_i - \gamma(f_i - 1) \tag{9}$$

$$\frac{df_i}{dt} = s_i \tag{10}$$

$$\tau_{MTT}\frac{dv_i}{dt} = f_i - v_i^{\frac{1}{\alpha}} \tag{11}$$

$$\tau_{MTT}\frac{dq_i}{dt} = f_i \frac{1 - (1-\rho)^{\frac{1}{f_i}}}{\rho} - v_i^{\frac{1}{\alpha}}\frac{q_i}{v_i} \tag{12}$$

where $\tau_{MTT}$ is the mean transit time of blood, $\rho$ is the resting oxygen extraction fraction, and $\alpha$ represents the resistance of the veins (i.e. stiffness). For each area ($i$), the BOLD signal ($B$), is a static nonlinear function of deoxyhemoglobin content ($q$) and venous blood volume (v), that comprises a volume-weighted sum of extravascular and intravascular signals:

$$B_i = V_0\left[k_1(1-q_i) + k_2\left(1 - \frac{q_i}{v_i}\right) + k_3(1-v_i)\right]$$
$$k_1 = 4.3\vartheta_0\rho TE$$
$$k_2 = \varepsilon r_0\rho TE \tag{13}$$
$$k_3 = 1 - \varepsilon$$

where $V_0$ is the resting venous blood volume fraction, $\vartheta_0$ is the frequency offset at the outer surface of the magnetized vessel for fully deoxygenated blood, $\varepsilon$ is the ratio of intra- and extra-vascular signals, $r_0$ is the slope of the relation between the intravascular relaxation rate $R_{2I}*$ and oxygen saturation, and $TE$ is the echo time of the fMRI acquisition. Parameters for the Balloon-Windkessel model matched those used previously for 3T fMRI experiments (*Stephan et al., 2007*). Simulated BOLD signals were downsampled to a temporal resolution of 2 s (i.e. TR = 2 s) to match the in vivo data and INT was estimated as for the in vivo data.

## Biophysical model analysis

To simulate the observed effects of hallucination and delusion severity on INT, we perturbed the strength of the couplings to the excitatory population from the local excitatory population ($w_{EE}$) or to the inhibitory population from the local excitatory population ($w_{IE}$) for specific nodes. Note that the original definition of the model assigned the same values of $w_{EE}$ and $w_{IE}$ to all nodes, but here we manipulated these values differentially across nodes. We investigated alterations in excitation-inhibition (E/I) ratios by allowing the strength of recurrent connections to vary in five of the six nodes that correspond to levels of our hierarchy (V1, V2, V4, MT, 8l, and 46d, with the latter being fixed) to

recapitulate our in vivo observations. Recurrent connection strength was fixed for 46d to avoid model instability upon small parameter changes (E/I ratio changes of ~1%) due to the strong connectivity at this level. The E/I ratio changes were modeled as a triangle function where a local maximum exhibited a peak E/I ratio increase and other nodes had E/I ratio changes that decreased linearly as a function of absolute distance in hierarchical levels from the peak. This function was described by three free parameters. (i) The hierarchical level of the peak E/I ratio increase, which was allowed to take any integer between 1 and 8. Given their stationary nature, these parameters were held constant such that fitting was performed for each combination of peak E/I ratio increase (1–8 for hallucinations and 1–8 for delusions) using a grid search. (ii) The magnitude of the E/I ratio increase at the peak (expressed as percent change to the local recurrent connection strength), which was allowed to vary between 0% and 40%. (iii) The magnitude of the E/I ratio change at the minimum (i.e. at the hierarchical level furthest from the peak), which was allowed to vary between -30% and 40%.

To facilitate fitting the biophysical model, we used regression fits from $M1_{primary}$ in the auditory system to estimate INT values at each level of the hierarchy for 4 'exemplary cases': (1) no hallucinations or delusions (fitted INT values from $M1_{primary}$ with minimum scores of 0 for both symptoms); (2) hallucinations only (maximum score of 5 for hallucinations and score of 0 for delusions); (3) delusions only (scores of 0 for hallucinations and 5 for delusions); (4) hallucinations and delusions (scores of 5 for both symptoms). For all exemplary cases, the severity of other symptoms and the values of covariates were set to the average values from all patients. Changes of INT for exemplary cases 2–4 were determined as the difference in INT relative to the 'no hallucinations or delusions' case (in vivo ΔINT). Model-derived in silico ΔINT were calculated for each node as the difference in INT from the unaltered biophysical model (i.e. $w_{IE}$ = 12.2 pA/Hz for all nodes). The parameters describing the E/I ratio changes were fit by minimizing the sum of squared errors between the in silico ΔINT (nodes: V1 [level 1], V2 [level 2], V4 [level 4], MT [level 5], 8l [level 8], and 46d [level 9]) and the in vivo ΔINT (parcels: A1 [level 1], LBelt [level 2], PBelt [level 4], RI [level 5], 8a [level 8], and 46 [level 9]). We simultaneously fit the three free parameters for each symptom (three parameters for hallucinations and three parameters for delusions) using in vivo ΔINT for exemplary cases 2–4 (18 data points) with the combined effect of hallucinations and delusions fit by the sum of E/I ratio changes for hallucinations and the E/I ratio changes for delusions. This was done by calculating the error between the biophysical model with E/I ratio changes for the hallucination parameters and exemplary case 2; the error between the biophysical model with E/I ratio changes for the delusion parameters and exemplary case 3; the error between the biophysical model with E/I ratio changes determined by the sum of the E/I ratio changes for the hallucination parameters and the E/I ratio changes for the delusion parameters, and exemplary case 4; and minimizing the sum of squared errors. Results are shown for reductions to $w_{IE}$, but similar effects were observed when increasing $w_{EE}$ since both effectively increase the E/I ratio.

## Acknowledgements

We thank Drs. Rishidev Chaudhuri and Xiao-Jing Wang for their guidance in implementing the large-scale biophysical model. We also thank Mr. Joshua Burt and Dr. John Murray for sharing their compilation of the Allen Human Brain Atlas data. This work was supported by the National Institute of Mental Health under awards R01MH117323 and R01MH114965. BrainGluSchi: data were downloaded from the COllaborative Informatics and Neuroimaging Suite Data Exchange tool (COINS; http://coins.mrn.org/dx) and data collection was funded by NIMH R01MH084898-01A1. COBRE: Data was downloaded from the COllaborative Informatics and Neuroimaging Suite Data Exchange tool (COINS; http://coins.mrn.org/dx), data collection was performed at the Mind Research Network, and funded by a Center of Biomedical Research Excellence (COBRE) grant 5P20RR021938/P20GM103472 from the NIH to Dr. Vince Calhoun. NMorphCH: data were obtained from the Neuro-morphometry by Computer Algorithm Chicago (NMorphCH) dataset (http://nunda.northwestern.edu/nunda/data/projects/NMorphCH); the investigators within NMorphCH contributed to the design and implementation of NMorphCH and/or provided data but did not participate in analysis or writing of this report; data collection and sharing for this project was funded by NIMH grant R01MH056584. UCLA: data was obtained from the OpenfMRI database (its accession number is ds000030) and data collection was funded by the Consortium for Neuropsychiatric Phenomics (NIH Roadmap for Medical Research grants UL1-DE019580, RL1MH083268, RL1MH083269,

RL1DA024853, RL1MH083270, RL1LM009833, PL1MH083271, and PL1NS062410). HCP: Data were provided by the Human Connectome Project, WU-Minn Consortium (Principal Investigators: David Van Essen and Kamil Ugurbil; 1U54MH091657) funded by the 16 NIH Institutes and Centers that support the NIH Blueprint for Neuroscience Research; and by the McDonnell Center for Systems Neuroscience at Washington University.

## Additional information

### Funding

| Funder | Grant reference number | Author |
|---|---|---|
| National Institute of Mental Health | R01MH117323 | Guillermo Horga |
| National Institute of Mental Health | R01MH114965 | Guillermo Horga |

The funders had no role in study design, data collection and interpretation, or the decision to submit the work for publication.

### Author contributions

Kenneth Wengler, Conceptualization, Data curation, Software, Formal analysis, Supervision, Validation, Investigation, Visualization, Methodology, Writing - original draft, Writing - review and editing; Andrew T Goldberg, Data curation, Software, Formal analysis, Investigation; George Chahine, Data curation, Software, Investigation; Guillermo Horga, Conceptualization, Resources, Supervision, Funding acquisition, Methodology, Writing - review and editing

### Author ORCIDs

Kenneth Wengler (iD) https://orcid.org/0000-0002-8153-5183
Guillermo Horga (iD) https://orcid.org/0000-0002-9049-9786

### Decision letter and Author response

Decision letter https://doi.org/10.7554/eLife.56151.sa1
Author response https://doi.org/10.7554/eLife.56151.sa2

## Additional files

### Supplementary files

• Transparent reporting form

### Data availability

All data analysed during this study are publicly available.

The following previously published datasets were used:

| Author(s) | Year | Dataset title | Dataset URL | Database and Identifier |
|---|---|---|---|---|
| Poldrack RA, Congdon E, Triplett W, Gorgolewski KJ, Karlsgodt KH, Mumford JA, Sabb FW, Freimer NB, London ED, Cannon TD, Bilder RM | 2016 | UCLA | https://openneuro.org/datasets/ds000030/versions/00016 | OpenNeuro, 10.12688/f1000research.11964.2 |

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
