## [Decision Letter]

**Acceptance summary:**

This works provides rigorous and novel insight into the brain mechanisms underlying psychosis, with distinct processes relating to hallucinations and delusions. The authors provide evidence for a hierarchical process in which alterations in the dynamics of somatosensory brain systems are related to delusions, whereas alterations in dynamics of auditory perceptual brain systems are related to hallucinations. Simulations from a computer model recapitulate these findings by altering the balance between excitation and inhibition in distinct hierarchical layers of a simulated circuit.

**Decision letter after peer review:**

[Editors’ note: the authors submitted for reconsideration following the decision after peer review. What follows is the decision letter after the first round of review.]

Thank you for submitting your work entitled "Distinct hierarchical alterations of intrinsic neural timescales account for different manifestations of psychosis" for consideration by *eLife*. Your article has been reviewed by three peer reviewers, including Claire M Gillan as the Reviewing Editor and Reviewer #1, and the evaluation has been overseen by a Senior Editor. The following individual involved in review of your submission has agreed to reveal their identity: Philip R Corlett (Reviewer #2).

Our decision has been reached after consultation between the reviewers. Based on these discussions and the individual reviews below, we regret to inform you that your work will not be considered further for publication in *eLife*.

As will be clear from the reviews, the reviewers agreed that the study posed an interesting, important and timely question, given much discussion in the field around abnormal hierarchical processing dysfunctions in schizophrenia. The premise was compelling and the use of relatively large pre-existing datasets and a new analysis methodology were all strengths. We each enjoyed reading it, but had similar reservations that led us to agree that this study would be better suited to a more specialist psychiatry journal. We think this work lays an important foundation for future research, which we suspect may require even larger samples to arrive at definitive conclusions.

In terms of the key factors that contributed to our decision the main result regarding the relationship between INT and auditory hierarchies was particularly striking, but all of the reviewers ultimately questioned the statistical robustness of the conclusion. This was due to a combination of factors that include (i) the somewhat arbitrary decision with respect to the ordering of regions within the auditory hierarchy, (ii) the fact that the result was marginal and would not survive some reasonable tests of alternative orderings within the hierarchy, (iii) that exploratory analyses appear to indicate effects of a similar magnitude for other symptoms of schizophrenia. The interpretation of the results with respect to delusions were perhaps less well-received, with multiple reviewers noting that results which failed to reach significance were, in parts, over-stated and over-discussed. The authors may like to take these opinions into account in a future submission to a more specialist journal, reducing the emphasis on the delusion result and presenting more fully the exploratory analyses so that future work can build on this excellent study in a more systematic way.

Reviewer #1:

This is a well-motivated and interesting paper that applies recently developed fMRI methods to study intrinsic neural timescales (INT) in a relatively large sample of schizophrenia patients. The findings are novel and I read it with great interest, the key result being that hallucinations are associated with an increase in INT at lower levels of the hierarchy of auditory cortex. Delusions did not show this pattern and trended towards the opposite, an increase in INT at higher levels of the hierarchy. The authors frame this in the context of recent theories of schizophrenia, where delusions are thought to arise from alterations in higher-order processing of information (concepts, beliefs, etc), while hallucinations are posited to stem from alternations in low-level stimulus processing. Again, I thought this was well-presented and I enjoyed reading it.

My key concern is that the results themselves are not 100% compelling. There is no statement about statistical power. All of the key effects are quite small and the significance levels are all just under p<.05. The key result is from auditory cortex, but there is clearly multiple testing (e.g. analyses at whole brain, and also in multiple sub-regions), but no correction has been applied beyond the permutation testing (which as I understand does not control for this), nor are stronger interaction tests (by brain region) carried out. Were a more strict criterion applied, results would not achieve significance. The exploratory analyses, which were not the focus of the study, indicate several other symptoms of schizophrenia that are associated with alterations in the gradient. Hallucinations was significant at p=.04, conceptual disorganisation p=.06 and blunted affect at p=.08 (the authors don't indicate direction). This casts some doubt over the specificity of these results, although I appreciate these are exploratory analyses that were in part predicated on seeing an opposing pattern or delusions. It would of course be more compelling to observe significant differences between hallucinations and all of the other symptoms.

That said, this is a new area and this paper serves as a nice foundational set of analyses for others to probe in the future. Should the authors be penalised for results that are more equivocal than they would have liked (or simply just a bit weaker)? Probably not. I suspect people will read with interest, we just need to ensure that the results are not over-stated.

Reviewer #2:

I read and enjoyed Wengler and colleagues' report of intrinsic resting state functional connectivity within sensory hierarchies and its relationship to hallucinations and delusions in patients with schizophrenia. They claim that hallucinations are related to perturbations lower in the hierarchy whereas delusions are related to higher hierarchical problems, in the auditory (and sensorimotor) but not visual hierarchies.

This is an important finding that may help to contextualize behavioural and computational findings that appear to show delusions relate to aberrant prediction errors (and apparently weak priors) and hallucinations to strong priors.

Whilst I am positively predisposed to this work, I have some concerns that I think should be addressed before publication.

1) Statistics. The claims the authors are making demand a significant omnibus f-test for the interaction between symptom (delusions vs hallucinations), system (visual, auditory, sensorimotor), vs level (high vs low). They report various components of this analysis as post-hoc t-tests, but no overall f-value rather t scores for some comparisons but not others, and the absence of significant effects for some comparisons, rather than testing the full interaction. If the overall comparison is significant, the unpacking will be appropriate and the result will be more believable.

2) Symptom contents. This may prove enlightening. There are some delusions that are more hallucination like, like delusions of parasitosis: the belief that one is infested with insects, which may be associated with tactile and visual hallucinations, where do people with these delusions fall on the hierarchical perturbations

3) Supplementary analyses. It is my understanding that *eLife* does not permit supplements. Why are supplements mentioned throughout? And why were some things relegated to the supplement? The sensorimotor analysis which is consistent with the auditory result is relevant to delusions of passivity too (and the apparent failures of corollary discharge/forward modeling that may under write them, this should be explored too if possible, are passivity delusions particularly related to changes in the sensorimotor hierarchy?

Furthermore, why did the authors exclude and then re-include DLPFC? It would seem very relevant to delusions from the lesion studies and some fMRI work.

Reviewer #3:

In this study, the authors use an approach first published by Watanabe et al., 2019, to estimate intrinsic neural timescales, INT (i.e. the rate of decay of the autocorrelation function) from resting state fMRI data in subjects with schizophrenia. They do this first in 100 healthy subjects from the HCP dataset, and find that INT can be reliably estimated from rsfMRI data, and that INT increases as one ascends the auditory and visual (and somatosensory) hierarchies. It doesn't have a clear relationship to other brain hierarchies (assessed using their T1w/T2w myelin content) however. They then analyse INT in some open schizophrenia datasets, and find INT is reduced globally in schizophrenia. They look at relations of INT gradient with hallucinations and delusions in the auditory and visual systems, and find that subjects with hallucinations have a positive relationship between INT and hallucinations in lower parts of the auditory hierarchy, despite their lower INT overall. There is a less convincing positive relationship between delusions and INT in the upper part of the auditory hierarchy. Neither is the case in the visual hierarchy. The authors go on to simulate these INT differences using a biophysical model, by increasing the self-connectivity in pyramidal cell populations more at the lower or higher ends of the hierarchy respectively.

This is an interesting paper and an important analysis to perform, given the widespread hypotheses about abnormal hierarchical message passing and pyramidal cell dysfunction in schizophrenia. The relationship between INT and auditory and visual hierarchies is striking. I do have some major reservations about some aspects of the paper, however:

1) My biggest reservation is that (unless I have misunderstood the statistics) the post-hoc test of the relative increase in INT at higher hierarchical levels in the auditory hierarchy in those with worse delusions is not significant (effect of hierarchy p=0.11). The actual p value for the delusion effect shown in Figure 3A seems to be given 32 pages later in the supplement (p=0.21). Yet the whole paper is framed around the hallucination and delusion effects. Really, all mention of any delusion effect should be removed from the paper, such an effect has not been found (unless I misunderstand, if so, many apologies). In addition, I find the motivation for the delusion effect far less persuasive than that for the hallucination effect (see below).

2) I am also not clear on to what extent the significance of the results depends on the strict order of areas given here. For example, what is the evidence that the auditory hierarchy is a linear progression from A1-LBelt-MBelt-PBelt-RI-A4-A5? To my (imperfect) knowledge the auditory hierarchy is complex and not well understood, it may contain two parallel hierarchies (e.g. Hackett, 2011, Hearing Research) and numerous regions are on the same “level” and thus could be listed in any order (e.g. Figure 1, Kaas and Hackett, 2000). Do the key results stand up to different reasonable permutations of the “hierarchical level” order in Figure 3A? Given the closeness of the p values to 0.05 I am concerned they would not…

3) I applaud the use of the simulations but I wonder how much they really add to the paper. In a sense it is a trivial result to show that increasing self-connection strength increases autocorrelation and hence INT: how could it not do so? Perhaps the simulations could be used to more closely match the size of the empirical effects, and thus estimate the rough order of magnitude of the possible changes in parameters that underlie them?

Some other points follow:

Introduction: the authors hypothesize that "INT at these respective levels would increase with more severe symptoms, reflecting increased neural integration of prior information". To me this prediction does not make sense with respect to delusions. From a neurophysiological point of view, I would expect intrinsic neural timescales as measured by these studies to reflect the ability to sustain neural activity, e.g. due to NMDAR function in pyramidal cells, or pyramidal interactions with interneurons, or network attractor dynamics: all of these processes are of the order of up to a few seconds. Delusions seem a different process entirely, likely encoded by long term synaptic plasticity? I don't see why they would have anything to do with INT? (Ongoing hallucinations on the other hand do fit this hypothesis).

Results and Figure 1: I don't understand why Figure 1E shows a mixture of best fit lines from a) 3 networks and b) 3 groups which have no network relationships i.e. “anterior”, “posterior” and “temporal”. What is the logic behind these latter groupings? Why not use other network groupings?

Subsection “Hierarchical Differences in Intrinsic Neural Timescales Between Hallucinations and Delusions”: I don't think the authors can interpret an effect at p=0.11 with any confidence, I would suggest removing the sentence about delusions being associated with an increase in higher level INT from the manuscript. The authors also refer to this "expanded hierarchical gradient related to delusions" elsewhere in the manuscript, e.g. in Figure 3B, 4, Discussion etc. I don't think this can be accepted as a finding, if they wish to include all the simulations then that is fine, but they should not be described as if referring to an empirical result.

In the Discussion the authors state "patients with more severe hallucinations exhibited a less pronounced hierarchical gradient, consistent with increased timescales at lower levels". Could this be rephrased to emphasise the timescales at lower levels are not increased relative to controls, but just that their gradient is more shallow?

In the Discussion the authors say "distinct hierarchical alterations provide symptom-specific pathways that together may explain symptom co-occurrence", but given that apparently opposite relationships exist between INT gradients and delusions vs hallucinations, how does this explain symptom co-occurrence? Would one not expect these symptoms to correlate negatively if these opposite relationships were correct and causal? Also in the next paragraph, the authors claim these findings "fit well with the timescale of symptoms", but this is not the case for delusions, for which it is hard to motivate a relationship with INT (as discussed above).

Apologies if I missed it but does post-hoc testing show a significant effect of hierarchy for hallucinations in the somatosensory system? Or is it just the interactions that are significant?

Materials and methods: Motion is clearly a concern given the authors have shown it is associated with reduced INT in the HCP sample. I see that motion scrubbing was performed, as well as a motion quality check, but was the motion in the schizophrenia group significantly higher than the control group even after these procedures? And if motion is used as a nuisance regressor, is the schizophrenia group still significantly associated with lower INT?

In the simulations, were w_EE_ in V1, V2 and V4 and also V2, V4 and MT increased by 10%, 5% and 2.5% respectively *in both cases*? Unless I misunderstand something there must be a misprint here, do you mean 2.5%, 5% and 10% for the latter set of areas?

In any case, are the effects in Figure 4 of the same order of magnitude as the effects observed in the fMRI data? The upper and lower hierarchy effects are also quite different to each other. Why not simulate effects of similar orders of magnitude to the detected effects, this would convey what magnitude of changes to these parameters might be needed to cause these pathologies… Also, I would have thought the most realistic simulations would in fact be ones in which w_EE_*decreased* throughout the hierarchy but in differing amounts depending on hierarchical level in the hallucination vs delusion cases. The simulation as it is has INT increasing above “normal” in the pathological cases, which is not what was observed in any area in the schizophrenia group, unless I'm mistaken?

[Editors’ note: further revisions were suggested prior to acceptance, as described below.]

Thank you for resubmitting your article "Distinct hierarchical alterations of intrinsic neural timescales account for different manifestations of psychosis" for consideration by *eLife*. Your revised article has been reviewed by two peer reviewers, and the evaluation has been overseen by Michael Frank as the Senior Editor and Reviewing Editor. The reviewers have opted to remain anonymous.

The reviewers have discussed the reviews with one another and the Reviewing Editor has drafted this decision to help you prepare a revised submission.

Overall the reviewers and I were impressed by your revision. As you will see however, there was variability in how convinced the reviewers were by the new results given that they depend on a new hierarchy. In the consultation session among reviewers, one reviewer noted that while the paper is innovating and exciting, they are troubled because they felt that links between brain and behavior should be tethered/grounded at both ends. They would like to be more sure that the new way that you have chosen to define the brain hierarchy isn't the one that happens to correlate with delusions, noting "would like to be convinced that there is independent validation of the hierarchy and that is indeed the one that we find in the brain, rather than the one that happens to work best for the authors' purpose".

The other reviewer was more convinced and thought your way of establishing the hierarchy via T1w/T2w and thickness was as good as you might get in humans, and that establishing this hierarchy would be a paper in itself, and that all 4 auditory rankings you obtained from the literature (without checking the primary sources) showed significant AVH effects and 3/4 significant delusion effects. So maybe the fine details of the middle-order ranking don't matter so much? In any case, looking at the Glasser et al., 2016, supplement, the myelin content fits the hierarchy you came up with. So all in all, this reviewer was fairly confident this auditory hierarchy is reasonable and not just picked for its data-fitting qualities.

They then followed this up noting that your winning visual hierarchy falls within the null distribution of model fits for myelin/thickness (Figure 1B). So it seems visual hierarchy is not very reliably measured at all. But the intrinsic timescale results weren't significant in the visual system either so that doesn't really matter. They noted "If anything I think they should stop treating all the sensory hierarchies similarly and point out the visual one seems quite different but this is a side issue".

Given these divergent opinions but with overall positive inclinations, I would like you to consider some more moderate way you could address this, e.g. by reporting more fully the AVH and delusion affects for all 4 auditory rankings and discussing the implications of the revised approach which then leads to relation to delusions.

Reviewer #2:

This revision and appeal is much improved.

It is challenging since we should not moderate our enthusiasm for a piece based on the specific results, however, the fact that the gradients now relate significantly and oppositely to hallucinations and delusions is encouraging.

Here is my remaining concern. The authors can't have it both ways. They reclassified the hierarchy and got this interesting and compelling pattern of findings. The pattern is even significant compared to a random ordering of regions. However, I would like to be reassured further that:

1) This is the most appropriate construction of hierarchy, i.e. the choice of hierarchy construction reflects biological reality (leveraging for example postmortem data on which there are also MRI data).

2) What impact the choice of hierarchy construction has on the symptom associations, that is, compared to some control other than random, how robust are the associations, given that they made some different choices and got a less robust set of effects.

To summarize, I would like to be more convinced that these effects are not being driven by the authors new choices about anatomy and hierarchy, and would like to be reassured that these are the best choices given what we know about the brain

Reviewer #3:

I think the authors have done a great job in responding to the comments and the paper is definitely stronger as a result. I have only a couple of comments.

I have some trouble understanding the new modelling part, the description is not clear in the text and neither in the figure legend. The different panels in Figure 4 are also not explicitly referenced in the text (at least not in the rebuttal letter). There is also not much labelling in Figure 4 itself. Could this all please be clarified? Some specific issues too:

The authors state "the best-fitting levels of the peak increase in local E/I ratio were levels 1 and 8" but this is a six node hierarchy? Should this be levels 1 and 6?

The in silico plots in Figure 4B look identical all along the row. Is that meant to be the case? I'm also not clear why in vivo auditory results are being compared with in silico visual ones?

The legend descriptions "insets for A" and "Insets for B" should be B and C respectively, I think?

Also in the phrase "Insets for A show predicted INT values" does “predicted” mean estimated from in vivo data? “Predicted” sounds like a model has been involved but I assume that is not the case? I don't understand the difference between the Insets for A and B?

---

## [Author Response]

[Editors’ note: The authors appealed the original decision. What follows is the authors’ response to the first round of review.]

In terms of the key factors that contributed to our decision, the main result regarding the relationship between INT and auditory hierarchies was particularly striking, but all of the reviewers ultimately questioned the statistical robustness of the conclusion. This was due to a combination of factors that include (i) the somewhat arbitrary decision with respect to the ordering of regions within the auditory hierarchy,

We agree with the reviewers that our justification for the ordering of regions within the auditory hierarchy was insufficient. While the ordering we chose was based on anatomy studies, it is true that there is no widely accepted ordering of the auditory hierarchy. To address this concern, we have performed an anatomically constrained model comparison of plausible models to determine the best ordering of regions in the auditory, visual, and somatosensory hierarchies. This was performed using myelin (T1w/T2w ratio) maps and cortical thickness maps – both widely used and validated structural measures of hierarchy – from high-quality data in 100 healthy HCP subjects. Furthermore, to also address a comment regarding the DLPFC (by the second reviewer) and to increase the range of the hierarchy (thereby improving statistical power), after determining the best order for the sensory cortex regions in the HCP dataset, we added two known downstream projections of the auditory, visual, and somatosensory cortices in the prefrontal cortex: area 8a and area 46. A second model comparison step was performed to determine the order of areas 8a and 46. Finally, we validated this hierarchy (determined via structural MRI measures of hierarchy) by showing that it significantly explains a substantial amount of variance in INT values (a functional MRI measure of hierarchy) in both the HCP subjects and, separately, on an external dataset (the 158 healthy control subjects in the combined schizophrenia dataset; both R^2^>0.63 for auditory system). Furthermore, we show that the explained variance of the chosen orders explains significantly more variance than random orderings. Thus, we now present a principled method for choosing the hierarchical ordering that we believe minimizes the need for arbitrary choices, since it is strongly rooted in primate anatomy and empirically validated via structural and functional human MRI measures. As an additional control analysis, we used an anatomically-agnostic definition of the hierarchy by performing a voxelwise analysis and determining the hierarchy by binning voxels based on INT values in the HCP data (with the lowest INT bin reflecting the lowest hierarchical level and the highest INT bin reflecting the highest hierarchical level). Results from this analysis are convergent with the main results (with the hierarchical gradients being significantly compressed for hallucinations and significantly expanded for delusions).

The following section has been added to the revised manuscript:

“Selection and Multimodal Validation of Neural Hierarchies

Our hypothesis of symptom-specific INT differences in hierarchical gradients was agnostic with respect to the specific neural hierarchies involved in psychosis, but involvement of most sensory modalities has been reported (Lewandowski et al., 2009; Postmes et al., 2014). […] Thus, we empirically validated extended sensory hierarchies that captured variability in structural and functional hierarchical indices across two independent samples, although this was surprisingly less evident for the visual system.”

See Figure 3—figure supplement 5.

(ii) the fact that the result was marginal and would not survive some reasonable tests of alternative orderings within the hierarchy,

We believe that our new method for empirical validation of the hierarchies circumvents the need for testing of alternative orderings, since we show that the selected auditory hierarchy captures a substantial amount of variability in structural measures of hierarchy (explaining over 80% of the variance) and in functional measures of hierarchy in two samples (explaining over 63% of the variance in each). Indeed, while the auditory hierarchy is less well established than the visual hierarchy, which is not generally seen as controversial, our winning hierarchical ordering for the auditory hierarchy explains considerably more variance in structural and functional MRI measures than that for the visual hierarchy. Therefore, we believe that the hierarchical ordering that we selected, even if imperfect, is a reasonable approximation to the true underlying hierarchy and thus allows us to test our hypothesis. That being said, all four of the auditory cortex orderings compared within the model comparison analysis (i.e., the orderings we considered to be most plausible given prior anatomical studies) showed significant negative hierarchical-gradient effects of hallucinations (all *P_permutation_* < 0.044) and all four showed significant (3 out of 4) or trend-level (1 out of 4) positive hierarchical-gradient effects of delusions (all *P_permutation_* < 0.064). Furthermore, an additional analysis shows that our results are significant when compared to a null distribution of randomly ordered auditory hierarchies (all *P_permutation_* < 0.001 for positive effect of delusions, negative effect of hallucinations, and their interaction), instead of compared to a null distribution of randomly permuted symptom scores.

(iii) that exploratory analyses appear to indicate effects of a similar magnitude for other symptoms of schizophrenia.

Using our newly defined auditory hierarchy including prefrontal cortex projections, we observe the strongest effects for positive symptoms, with hallucinations (t = -5.51 , *P_permutation_* = 0.005) being the strongest negative effect and delusions (t = 3.12, *P_permutation_* = 0.031) being the strongest positive effect; conceptual disorganization (t = -3.19, *P_permutation_* = 0.026) is the only other significant effect observed. All other symptoms are associated with non-significant effects *P_permutation_* > 0.11. Thus, all three positive symptoms (including disorganization) and none of the negative symptoms show significant effects, which we take to reflect some level of selectivity that we now examine in more detail. It is important to note that the perceptual-inference model of psychosis we set out to test does not require these effects to be specific to hallucinations and delusions. Also, we don’t believe that an effect of disorganization (a positive symptom that unlike negative symptoms tends to correlate with hallucinations and delusions) provides a challenge for the hypothesized model – indeed it may suggest extensions of the model to account for additional phenomena. Finally, we chose to include the analyses of other symptoms for completeness since this is the first study to investigate INT in schizophrenia, hoping the exploratory analysis results would provide interesting future directions. But our hypothesis was exclusively focused on hallucinations and delusions. This is partly because we reasoned that finding distinct correlates of symptoms that tend to correlate would provide a stringent test, the results of which would be most informative for the hypothesized model of psychosis.

To elaborate on this point, the following has been added to the revised manuscript:

“Post-Hoc Analysis on the Specificity of INT Hierarchical-Gradient Effects

In a post-hoc analysis, we then investigated the specificity of these hierarchical-gradient effects to the positive psychotic symptoms under investigation. […] Thus, although the hierarchical-gradient effects were not unique to the two symptoms under investigation – which is not to be required under perceptual-inference models of psychosis and which could suggest model extensions to account for additional phenomena – these effects were strongest for, and relatively specific to, positive symptoms.”

The interpretation of the results with respect to delusions were perhaps less well-received, with multiple reviewers noting that results which failed to reach significance were, in parts, over-stated and over-discussed.

We agree that our results for delusions were over-stated and over-discussed. Our intention was to emphasize that based on the perceptual-inference model of psychosis, the critical test was the interaction between hallucinations and delusions. And since we showed a significant effect for hallucination severity that was significantly moderated by delusion severity, we took this to support the model, which we wanted to elaborate upon in the discussion (which, partly due to its complexity, could not be fully described in the Introduction). Furthermore, the results for delusions were significant in some of the control analyses and in all analyses using a standard test not based on permutations, so we thought of it as a trend. But we realize the rationale was unclear. That said, this should be less problematic now, since the analysis using the newly defined hierarchy yields significant main effects for both delusions and hallucinations as well as a significant interaction effect (all via a stringent permutation test). This is also true in the control analysis using INT-based bins instead of an anatomically informed ordering (see Figure 3—figure supplement 5). Furthermore, we also observe a significant hierarchical-gradient effect for delusions in the somatosensory cortex.

Reviewer #1:[…]My key concern is that the results themselves are not 100% compelling. There is no statement about statistical power. All of the key effects are quite small and the significance levels are all just under p<.05.

We thank the reviewer for raising this concern. To address this, we conducted an anatomically constrained data-driven model selection of neural hierarchies. By including PFC downstream regions of sensory cortex, again informed by known anatomical projections and supported by empirical proxy measures of structural hierarchy (myelin content and cortical thickness; see above), we aimed to augment the hierarchical range under study and thereby increase sensitivity and statistical power for our a priori hypothesis testing. Following this approach, we specifically added two additional levels to the hierarchies that are downstream projections for both the auditory and visual cortices, areas 8a and 46 (the latter being particularly relevant to delusions and psychosis based on Corlett et al., 2007). Although the P-values for some of the key effects are still not very small (*P_permutation_* = 0.006, 0.029, and 0.045 for hierarchical interaction effect between hallucinations and delusions, hierarchical effect for hallucinations, and hierarchical effect for delusions, respectively, for the auditory hierarchy), the effect sizes fall within the large range (Cohen’s *f^2^* = 1.00, 0.41, and 0.27, respectively). Particularly, the interaction effect, which we believe is key to testing our hypothesis (as it provides statistical support that gradient effects differ between hallucinations and delusions), is consistently significant across all analyses and well below the significance threshold. A post-hoc power analysis of the effect sizes suggests >96% power for these observed effects (a = 0.05). In addition, the tests we use here are based on a non-parametric permutation test that is substantially stricter than a standard parametric test (i.e., all the relevant p values are substantially smaller for standard parametric significance tests). Furthermore, we hope that the reviewer finds our main results more compelling in light of the evidence against vibration of effects (i.e., different control models and analysis showing similar results) and our converging findings using a purely data-driven approach to determine auditory hierarchies (based on INTbased bins). Additionally, and as recommended in previous work dealing with issues of statistical power and confidence in observed results (Button et al., Nature Reviews Neuroscience 2013), we now report bootstrapping-based confidence intervals for significant effects. A final consideration is that we show that the INT measure has excellent test-retest reliability, substantially higher than for many other widely used fMRI-based measures (Plichta et al., NeuroImage, 2012; Birn et al., NeuroImage, 2013; Noble et al., Cerebral Cortex, 2017; Choe et al., NeuroImage, 2017; Zhang et al., NeuroImage, 2018), a factor that contributes to increasing the statistical power (and decrease the required sample size) of tests involving this measure (Zuo, Xu and Milham, Nature Human Behavior, 2019).

In response to this comment, we added measures of effect size for all main analyses and the following statement about statistical power:

“A post-hoc power analysis for M2 showed our analyses had between 88% and 99% power to detect effect sizes (Cohen’s f^2^) between 0.19 and 0.36 (a = 0.05).”

The key result is from auditory cortex, but there is clearly multiple testing (e.g. analyses at whole brain, and also in multiple sub-regions), but no correction has been applied beyond the permutation testing (which as I understand does not control for this), nor are stronger interaction tests (by brain region) carried out. Were a more strict criterion applied, results would not achieve significance.

We thank the reviewer for the opportunity to clarify our analysis plan. We did run a number of analyses (e.g. analyses for validation of hierarchies, which we take as a selection step, or exploratory analyses of diagnostic differences and of potential confounds) but we would like to clarify that our main, a priori analysis is that testing for differences in hierarchical INT effects for hallucinations and delusions (within the second-level model testing hierarchy-by-symptom-by-sensory-system effects). Within this model, we do test for hierarchical gradient symptom effects and differences by symptom in 3 sensory systems. We chose to use a single model that included all effects of interest (hierarchy-by-symptom-by-sensory-system). Within this model, individual parametric tests of regression coefficients account for multiple comparisons by adjusting the degrees of freedom (with more complex models effectively increasing the threshold for significance), thus guarding against false positives. However, this is only true for the parametric t-tests, which despite this correction we found to be too lenient (note that all positive effects reported here were significant within the model after this adjustment of degrees of freedom in the t-test for individual regression coefficients). Since we chose to use a permutation test of individual regression coefficients that is more stringent but formally lacks this adjustment, in response to this point we have added a family-wise correction using permutation test to further account for potential false positives. Here, we determined the chance level of observing the set of significant effects we report (i.e., at least 2 interaction effects of hierarchy-by-symptom, 1 negative effects of hierarchy for hallucination severity, and 2 positive effect of hierarchy for delusion severity, all consistent with our hypothesis) in the context of all the tests we run for coefficients within the main model (i.e., one test for each of the 2 symptoms plus one interaction test for each of 3 systems, for a total of 9 tests). This test shows that our results (with 5 out of 9 tests showing significant effects in the expected direction) indeed survive this family-wise-error correction. The following was added to the Results section of the revised manuscript:

“To correct for multiple comparisons, we carried out a family-wise permutation test determining the probability of spuriously obtaining the set of significant a priori effects we observed in support of our original hypothesis. […] Furthermore, based on the chance level of observing a significant negative hierarchical-gradient effect for hallucinations, and a significant positive hierarchical-gradient effect for delusions, and a significant symptom-by-hierarchical-level interaction (i.e., all 3 effects in one system), this analysis suggested that the observed set of results in the auditory system was also statistically above chance (set-level P_permutation_ = 0.043).”

The exploratory analyses, which were not the focus of the study, indicate several other symptoms of schizophrenia that are associated with alterations in the gradient. Hallucinations was significant at p=.04, conceptual disorganisation p=.06 and blunted affect at p=.08 (the authors don't indicate direction). This casts some doubt over the specificity of these results, although I appreciate these are exploratory analyses that were in part predicated on seeing an opposing pattern or delusions. It would of course be more compelling to observe significant differences between hallucinations and all of the other symptoms.

Please see our response to the third major concern from the editorial summary.

Reviewer #2:[…]1) Statistics. The claims the authors are making demand a significant omnibus f-test for the interaction between symptom (delusions vs hallucinations), system (visual, auditory, sensorimotor), vs level (high vs low). They report various components of this analysis as post-hoc t-tests, but no overall f-value (rather t scores for some comparisons but not others, and the absence of significant effects for some comparisons, rather than testing the full interaction. If the overall comparison is significant, the unpacking will be appropriate and the result will be more believable.

We thank the reviewer for pointing this out; we apologize for omitting this from the original submission. The omnibus f-test in the main model (and other control models) is significant, which thus justifies the main tests of interactions by symptom that we were primarily interested in. We also made an effort to report relevant statistics more systematically and comprehensively throughout the manuscript. (Note also that the graphs in Figure 3B provide complete t statistics for each term in the main regression model, which we show with alternative coding of reference categories to parse out the main effects. If the reviewer thinks it would be clearer, we could also add a supplemental table with complete statistics.) Regarding the omnibus test, this has been added to the Results section of the revised manuscript:

“The model explaining symptom effects and their differences by hierarchical-level and their interaction by symptoms and sensory system was significant (omnibus F_11,41_ = 5.52, P < 10^-4^).”

2) Symptom contents. This may prove enlightening. There are some delusions that are more hallucination like, like delusions of parasitosis: the belief that one is infested with insects, which may be associated with tactile and visual hallucinations, where do people with these delusions fall on the hierarchical perturbations

We agree with the reviewer that this is a very interesting question and is something we are looking into exploring in the future. Unfortunately, we do not have the data to answer this question with sufficient confidence. The SAPS scale was only available for roughly half of our subjects (*N* = 56) and a comprehensive investigation of this question would require a very large dataset. That said, here we present exploratory analyses using the SAPS scale in this smaller subset of subjects, which allowed for a more fine-grained investigation of symptom content. Specifically, we performed an additional analysis where delusion (or hallucination) severity was replaced with the score of one delusion (or hallucination) SAPS subitem in the firstlevel model. The strengths of these effects were then carried to the second-level model to determine their hierarchical effects (as in our main analyses) and these effects were then ranked to qualitatively determine if certain hallucination or delusion subitems could be driving the hierarchical effects we observed for the parent symptom. We observed some qualitative effects that we would have expected a priori: e.g., auditory hallucinations had the strongest negative hierarchical-gradient effect in the auditory system (with this symptom modality thus being likely to drive the observed gradient effect in the corresponding system), somatic hallucinations had the strongest negative hierarchical-gradient effect in the somatosensory system (with the symptom modality again matching the system), and delusions of being controlled had the strongest positive hierarchical gradient effect in the somatosensory system (consistent with corollary discharge and related models). Nonetheless, there are a number of effects that are more counterintuitive and some that showed opposite effects to the observed hierarchical-gradient effects of hallucinations and delusions in general. Thus, we believe that these results are difficult to interpret. We again agree with the reviewer that this type of analysis would be of great interest. But due to the limited sample size with sufficiently fine-grained clinical information, we believe our study is not well suited to address questions of symptom content with sufficient precision. Beyond the sample size, our confidence in this analysis is particularly low given the skewed distribution of symptom scores for the delusion and hallucination SAPS subitems, with most patients having a score of 0 for many individual subitems. Thus, we believe that a meaningful answer to this question would require a very large sample that ensures a wide range of severity in each subitem, which was clearly not the case in the subsample with available SAPS data. In relation to this important point, the following text has been added to the discussion as a future direction:

“Furthermore, our study was limited to investigating the effects of global severity of hallucinations and delusions and could not resolve effects of symptom subtype or content, since detailed assessments were only available in a small subset of our patients. Larger studies with more detailed assessments are needed to tease out these potential effects.”

3) Supplementary analyses. It is my understanding that eLife does not permit supplements. Why are supplements mentioned throughout? And why were some things relegated to the supplement? The sensorimotor analysis, which is consistent with the auditory result is relevant to delusions of passivity too (and the apparent failures of corollary discharge/forward modeling that may under write them, this should be explored too if possible, are passivity delusions particularly related to changes in the sensorimotor hierarchy?

Figure supplements are allowed and encouraged by *eLife*. While traditional supplemental information is discouraged, we did include 3 supplements: Methods, Exploratory Analyses, and Discussion. We have moved elements of the original supplemental discussion to the main text but would like to keep the supplemental methods as a supplement due to their length (mainly regarding the biophysical model, as we believe is important to give specific details on implementation that do not fit in the main text). That said, we will remove those sections, and perhaps add some more detail in figure captions, if the reviewer and the editors deem it appropriate. As for the Exploratory Analyses, these were relegated to the supplement because they do not directly pertain to our hypothesis regarding hierarchical perceptual-inference models of psychosis and we did not want them to distract readers from the goal of the paper. We have, however, included a figure depicting the hierarchical-gradient effects of all symptoms into the main text to address concerns regarding specificity of effects. Furthermore, given the relevance of the somatosensory system to psychosis, we have included it in all main analyses throughout the manuscript. With regard to the relevance of the somatosensory system to passivity delusions, we found some evidence for this but are not confident about these results for the reasons explained above in response to Comment 2.

Furthermore, why did the authors exclude and then re-include DLPFC? It would seem very relevant to delusions from the lesion studies and some fMRI work.

Thank you for pointing this out. Originally we did not include the DLPFC in our main text because of difficulties in assigning an exact hierarchical level to it (although it is well known from tract tracing studies that DLPFC is a downstream region receiving projections from the highest-level regions within auditory, visual, and somatosensory cortices). In response to this suggestion and the first reviewer’s comment about statistical power, however, we have now chosen to include the DLPFC (area 46) and area 8a – both downstream targets of auditory, visual, and somatosensory cortices (Felleman and Van Essen, 1991; Kaas and Hackett, 2000) – to our hierarchies in order to expand the covered range of these hierarchies, refrain from excluding relevant anatomical regions to delusions, and to increase statistical power (see response for point 1 in the editorial summary for further details).

Reviewer #3:[…]1) My biggest reservation is that (unless I have misunderstood the statistics) the post-hoc test of the relative increase in INT at higher hierarchical levels in the auditory hierarchy in those with worse delusions is not significant (effect of hierarchy p=0.11). The actual p value for the delusion effect shown in Figure 3A seems to be given 32 pages later in the supplement (p=0.21). Yet the whole paper is framed around the hallucination and delusion effects. Really, all mention of any delusion effect should be removed from the paper, such an effect has not been found (unless I misunderstand, if so, many apologies). In addition, I find the motivation for the delusion effect far less persuasive than that for the hallucination effect (see below).

Please see our response to the fourth point from the editorial summary.

2) I am also not clear on to what extent the significance of the results depends on the strict order of areas given here. For example, what is the evidence that the auditory hierarchy is a linear progression from A1-LBelt-MBelt-PBelt-RI-A4-A5? To my (imperfect) knowledge the auditory hierarchy is complex and not well understood, it may contain two parallel hierarchies (e.g. Hackett 2011 Hearing Research p138) and numerous regions are on the same “level” and thus could be listed in any order (e.g. Figure 1, Kaas and Hackett, 2000). Do the key results stand up to different reasonable permutations of the “hierarchical level” order in Figure 3A? Given the closeness of the p values to 0.05 I am concerned they would not…

Please see our response to the first and second points from the editorial summary.

3) I applaud the use of the simulations but I wonder how much they really add to the paper. In a sense it is a trivial result to show that increasing self-connection strength increases autocorrelation and hence INT: how could it not do so? Perhaps the simulations could be used to more closely match the size of the empirical effects, and thus estimate the rough order of magnitude of the possible changes in parameters that underlie them?

We thank the reviewer for this insightful comment and suggestion, which encouraged us to more carefully evaluate the model in question and its implications. We chose to include the simulations because we believe they are helpful in suggesting biologically plausible mechanisms that could be more directly examined in the future. While we agree that our simulation showing that increasing self-connection strength increases INT may seem trivial to those with a working knowledge of biophysical models, we do not believe this to be apparent to a general readership including non-modelers. Furthermore, because we use a large-scale network that captures brain-wide hierarchies which features short- and long-range connections of varying strengths between every node (unless no connections exist anatomically), changing the self-connection strength in one node affects all connected nodes. This model is thus more complex than a single-layer model where increasing the self-connection strength will always lead to a proportional local increase in INT, and we thus felt that actually showing the effects of manipulating the large-scale biophysical model was informative. That said, we agree that a more comprehensive examination of the model makes it even more valuable. Therefore, we have followed the reviewer’s suggestion and now directly fit the biophysical model to the estimated changes in INT as a function of symptom severity of hallucinations and delusions (from the first-level GLM; M1_primary_). We do this for different relevant clinical profiles: hallucinations only, delusions only, and both hallucinations and delusions. Because of the large number of possible free parameters, and current working models implicating changes in E/I ratio, we use 3 parameters for each symptom to describe a gradient of changes to self-connection strengths of 5 nodes in the hierarchy during fitting. Similar to our previous results, we observe a gradient of alteration in E/I ratio for hallucinations (larger increase in E/I ratio at low levels than at high levels) and for delusions (larger increase in E/I ratios at high levels than at low levels). These parameters were determined relative to an unaltered model. (Note that our focus was on modeling changes in INT comparing highly symptomatic versus asymptomatic patients and that using an unaltered model or using an altered model reflecting a schizophrenia phenotype leads to equivalent results; we thus chose the unaltered model as reference for simplicity and clarity). Our results also show that the effect of both delusions and hallucinations could be caused by a combined effect of the individual low-level effects of hallucinations and high-level effects of delusions resulting in increased E/I ratio throughout the hierarchy. The following was added to the revised manuscript:

“Altered E/I Ratio as a Potential Biological Mechanism

To explore candidate biological mechanisms for the effects we observed in vivo, we leveraged a large-scale biophysical model previously shown to capture intrinsic timescale hierarchies (Chaudhuri et al., 2015). […] Although preliminary, these results provide some support for the notion of additive hierarchical alterations underlying hallucinations and delusions.”

“Large-Scale Biophysical Model of Cortical Neural Timescales

We used a computational model of macaque cortex previously shown to capture the hierarchy of neural timescales observed using electrophysiology (Chaudhuri et al., 2015). […] A linear gradient in E/I ratio change was assumed between the peak and the minimum that was symmetrical around the peak.”

Some other points follow:Introduction: the authors hypothesize that "INT at these respective levels would increase with more severe symptoms, reflecting increased neural integration of prior information". To me this prediction does not make sense with respect to delusions. From a neurophysiological point of view, I would expect intrinsic neural timescales as measured by these studies to reflect the ability to sustain neural activity, e.g. due to NMDAR function in pyramidal cells, or pyramidal interactions with interneurons, or network attractor dynamics: all of these processes are of the order of up to a few seconds. Delusions seem a different process entirely, likely encoded by long term synaptic plasticity? I don't see why they would have anything to do with INT? (Ongoing hallucinations on the other hand do fit this hypothesis).

We thank the reviewer for raising this important point. While we do not necessarily argue against the view that delusions may require an additional involvement of qualitatively different processes like long-term synaptic plasticity changes contributing to their maintenance, our hypothesis of a unifying mechanism for psychosis is motivated by converging lines of theoretical and empirical work. These include (1) theoretical models of psychosis which aim to explain hallucinations and delusions as resulting from altered inference, (2) initial empirical support from our group and others involving altered inference at the timescale of sensory events occurring on the order of a few hundred milliseconds to a few seconds in the pathophysiology of both hallucinations and delusions, (3) multiple clinical studies showing that hallucinations and delusions tend to selectively cluster, across and within subjects (e.g. Breier and Berg, Biological Psychiatry, 1999), suggesting a common syndromal mechanism underlying both symptoms, (4) converging work suggesting excess striatal dopamine transmission underlying hallucinations and delusions (but not necessarily other aspects of the illness) and non-dopaminergic pharmacological manipulations that can induce both symptoms (e.g. ketamine; Corlett, Honey, and Fletcher, 2016; Corlett et al., 2011), and (5) clinical trials showing beneficial effects of antidopaminergic drugs on psychotic symptoms, including hallucinations and delusions (Breier and Berg, Biological Psychiatry, 1999), with a similar time course of response across these symptoms (Gunduz-Bruce et al., American Journal of Psychiatry, 2005). Given this, we believe hierarchical-inference models of psychosis have the advantage that they are more parsimonious than other views where hallucinations and delusions may results from completely different mechanisms (which in principle would not explain their clustering, common underlying neurobiology, or common response to pharmacotherapy) and have the flexibility to accommodate inferential subprocesses which are distinct but interdependent with subprocesses at other levels of the hierarchy. Note that other integrative models of schizophrenia (e.g. Maia and Frank, Biological Psychiatry, 2017), which similarly aim to provide a parsimonious account of the psychotic syndrome, also assume that hallucinations and delusions arise from a shared mechanism (e.g. related to dopamine alterations) rather than from completely different mechanisms. Thus, we believe that assuming a shared mechanism, even one that affecting distinct but interdependent levels of processing, is an important constraint to ensure a parsimonious explanation (in line with Corlett, Frith, and Fletcher, 2009; Adams et al., 2013; Horga and Abi-Dargham, 2019).

We nonetheless agree that it may be difficult relating changes in timescales of the observed magnitude to delusions, which are likely to occur, or at least be maintained, at a much longer timescale. However, there are several considerations that suggest a possible reconciliation. First, it is not unlikely that delusions arise within a relatively short timescale – e.g. in line with phenomenological descriptions of “delusional perception” or the “apophenia” that marks the beginning of the delusional process during the initial stages of psychosis described by Klaus Conrad and others – but persist over much longer timescales, similar to salient events or realizations occurring at a particular point in life that can be remembered for decades. Indeed, our previous work suggests that an alteration in higher-level inferences on hidden states which we measured in the lab within the timescale of seconds (and is thus unlikely to depend primarily on long-term plastic changes) selectively correlates with delusional severity in schizophrenia (Baker et al., 2019). This work indeed suggests a primacy bias in belief updating as a possible neurocognitive candidate for delusions, a bias that leads to excessive weighting of prior information and that could thus exaggerate the influence of older beliefs over time, making them persist over longer time periods. We take this to mean that the relevant timescale for altered inferences in relation to delusions may be the relevant timescale for integrating across events or information samples relevant to a given inferential process (for instance, when trying to infer on someone’s intentions by integrating information from different statements within a single conversation), and which in some cases may be in the order of seconds. Sometimes the inferential process of evidence integration will occur over longer timescales (e.g. days) but it may be already biased by integration of events close in time (e.g. separated by seconds), or may be rehashed after a memory-retrieval step that need not itself be altered. One could readily simulate an inferential process of evidence accumulation taking place within a minutes-long session (integrating several events close in time) and across daily or monthly sessions (integrating the stored inferences from previous sessions with new events in a separate session) to show that within-session biases occurring at a shorter timescale would affect inferences over a longer timescale, assuming that final inferences from one session are retrieved and used as a starting point for evidence integration in the next session (even when assuming some memory decay or noise). Given this theoretical argument and our prior empirical work, we thus believe that the seemingly short timescale we evaluate here is likely relevant to delusions.

Furthermore, it is important to note that the higher-level regions whose “resting” or “intrinsic” timescale (resting-state INT) is in the order of a few seconds, may substantially increase during processes engaging these regions. This is supported by results from simulations using the biophysical model from Chaudhuri et al., 2015, who showed changes in timescales as a function of stimulation, and by fMRI work showing substantially longer timescales measured under stimulation (Hasson et al., 2008). Thus, the small absolute differences in INT across levels of the hierarchy or as a function of symptom severity, may reflect a circuit-level alteration identifiable in spontaneous activity but which translates into substantially larger timescale differences in a system engaged in inferential processes.

Finally, the reviewer is completely correct in assuming that INT depends on sustained activity and the factors that may control it, such as NMDAR function, or pyramidal interactions with interneurons, or network attractor dynamics. But in contrast to the reviewer’s suggestion, these factors have been repeatedly linked to delusions in the literature. In particular, attractor models have been invoked to explain delusions (Chen, Canadian Journal of Psychiatry, 1994; Adams et al., Journal of Neuroscience, 2018), and that these models have been linked to NMDAR function (Loh, Rolls, and Deco, Pharmacopsychiatry, 2007).

For these reasons, we would argue that we have a strong rationale to study hallucinations and delusions within a hierarchical framework assuming alterations at different yet interconnected hierarchical levels. Similar frameworks have been successfully used to explain temporal nesting of information processing during speech in theoretical and empirical work: from the lower level of processing of syllable sounds, which require processing at a fast timescale, to higher levels related to semantic contexts and conceptual beliefs, which like delusions can be argued to persist over very long time periods (and which likely involve longer-term memory as well). We thus believe that, while many aspects will require further examination (e.g. the role of long-term memory processes), this framework provides a promising integrative mechanism to explain various aspects of the psychotic syndrome within a unified framework.

The following was added to the revised manuscript for clarification:

“One outstanding question is how the alterations in neural timescales we observed here in relation to delusions, which may predominate in high levels of the hierarchy yet manifest as changes on the order of seconds, may drive delusions evolving over much longer timescales. […] Although less critical, it is also worth noting that INT reflects differences in resting circuit dynamics, the timescale of which is likely to be substantially magnified when these circuits are engaged (Chaudhuri et al., 2015; Hasson et al., 2008).”

Results and Figure 1: I don't understand why Figure 1E shows a mixture of best fit lines from a) 3 networks and b) 3 groups which have no network relationships, i.e. “anterior”, “posterior” and “temporal”. What is the logic behind these latter groupings? Why not use other network groupings?

These groupings were defined by Glasser et al. in their parcellation paper. The “anterior”, “posterior”, and “temporal” groups are defined in such a way because many of these parcels are involved in multiple networks, making it difficult to separate them into network groupings. To better investigate the T1w/T2w-INT relationship at a network level, we have included the same figure showing parcel groupings according to the Cole-Anticevic Networks. The results are similar to the original groupings, where a model including separate intercepts and slopes for each network better describes the data, again supporting the lack of a universal relationship between T1w/T2w and INT across the whole brain. This is in line with the Chaudhuri et al. biophysical model, which suggests that disproportionate stimulation of a given sensory system (e.g. more auditory than visual stimulation during a resting-state fMRI scan) could exaggerate INT hierarchical gradients in the more stimulated system and change the pattern of INT across different regions. See Figure 1—figure supplement 2.

Subsection “Hierarchical Differences in Intrinsic Neural Timescales Between Hallucinations and Delusions”: I don't think the authors can interpret an effect at p=0.11 with any confidence, I would suggest removing the sentence about delusions being associated with an increase in higher level INT from the manuscript. The authors also refer to this "expanded hierarchical gradient related to delusions" elsewhere in the manuscript, e.g. in Figure 3B, 4, Discussion etc. I don't think this can be accepted as a finding, if they wish to include all the simulations then that is fine, but they should not be described as if referring to an empirical result.

We agree that our results for delusions were over-stated and over-discussed. Our intention was to emphasize that based on the perceptual-inference model of psychosis, the critical test was the interaction between hallucinations and delusions. And since we showed a significant effect for hallucination severity that was significantly moderated by delusion severity, we took this to support the model, which we wanted to elaborate upon in the discussion (which, partly due to its complexity, could not be fully described in the Introduction). Furthermore, the results for delusions were significant in some of the control analyses and in all analyses using a standard test not based on permutations, so we thought of it as a trend. But we realize the rationale was unclear. That said, this should be less problematic now, since the analysis using the newly defined hierarchy yields significant main effects for both delusions and hallucinations as well as a significant interaction effect (all via a stringent permutation test). This is also true in the control analysis using INT-based bins instead of an anatomically informed ordering (see Figure 3—figure supplement 3 above). Furthermore, we also observe a significant hierarchical-gradient effect for delusions in the somatosensory cortex.

In the Discussion the authors state "patients with more severe hallucinations exhibited a less pronounced hierarchical gradient, consistent with increased timescales at lower levels". Could this be rephrased to emphasise the timescales at lower levels are not increased relative to controls, but just that their gradient is more shallow?

Thank you for pointing this out, we agree that the suggested phrasing better reflects the results. For clarification, we have rephrased this in the revised manuscript as:

“Patients with more severe hallucinations exhibited a less pronounced hierarchical gradient, consistent with increased timescales at lower levels compared to those with less severe hallucinations, and those with more severe delusions instead exhibited a more pronounced hierarchical gradient, consistent with increased timescales at higher levels compared to those with less severe delusions (Figure 3c).”

In the Discussion the authors say "distinct hierarchical alterations provide symptom-specific pathways that together may explain symptom co-occurrence", but given that apparently opposite relationships exist between INT gradients and delusions vs hallucinations, how does this explain symptom co-occurrence? Would one not expect these symptoms to correlate negatively if these opposite relationships were correct and causal? Also in the next paragraph, the authors claim these findings "fit well with the timescale of symptoms", but this is not the case for delusions, for which it is hard to motivate a relationship with INT (as discussed above).

We thank the reviewer for allowing us to clarify this point. Our hypothesis was that hallucinations and delusions would predominantly affect different levels of neural hierarchies, which would manifest as distinct changes in the observed hierarchical gradients consistent with our observations. But given that one underlying mechanism for this local increases in E/I in low or high levels for hallucinations or delusions, respectively, these local alterations could combine in an additive manner: i.e., a patient could have increased E/I at both levels and present with both symptoms. If this were the case, as we now show in the extended simulations of the biophysical model, rather than canceling out, the combination of low- and high-level alterations would lead to increased INT across the levels of the hierarchy, with the hierarchical gradient being flatter but its intercept being higher. Thus, this model suggests that patients may exhibit changes in the hierarchical INT gradients if one of the positive symptoms predominates (with hallucinations flattening the gradient and delusions making it steeper) and with overall increases in INT throughout the hierarchy if both symptoms are severe. Our data is consistent with this: (1) the average INT in all 9 auditory parcels was significantly higher for subjects with both symptoms compared to those with neither; and (2) a mixed-effects model estimating the hierarchical effect in the auditory system (INT ~ 1 + hierarchy) showed a significantly larger intercept for subjects with both symptoms compared to those with neither. Give this, we argue that similar mechanisms in partially separable pathways may be responsible for hallucinations and delusions. Since both effects can be additive, and since both could be caused by a common underlying alteration, this framework accommodates clinical presentations featuring one symptom in isolation or their co-occurrence. Future work is of course needed to uncover the common underlying alterations and further elaborate on this model, but we think this is an important first step. In particular, the model we use is not “trained” or “optimized” to perform a certain task via Hebbian plasticity rules (unlike models such as Soltani and Wang, Nature Neuroscience, 2010). We envision that a biophysically realistic model trained to perform relevant tasks with hierarchical structure will likely feature partially related changes in plasticity at different hierarchical levels, thus potentially providing an explanation for why a single alteration in inferential circuits will tend to simultaneously affect low and high hierarchical levels. But this is beyond the scope of the current work. The following has been added to the revised manuscript:

“Importantly, an additive combination of the low-level (hallucinations) and the high-level (delusions) changes closely approximated the combined effect of hallucinations and delusions, which consisted of a general increase of INT with no clear change in the hierarchical gradient. […] Although preliminary, these results provide some support for the notion of additive hierarchical alterations underlying hallucinations and delusions.”

Apologies if I missed it but does post-hoc testing show a significant effect of hierarchy for hallucinations in the somatosensory system? Or is it just the interactions that are significant?

We have included the somatosensory system in all main analyses of the revised manuscript. The results are now reported more clearly, and we find a significant effect of hierarchy for delusions (in the same direction as in the auditory system) as well as a significant interaction of hierarchy between hallucinations and delusions.

Materials and methods: Motion is clearly a concern given the authors have shown it is associated with reduced INT in the HCP sample. I see that motion scrubbing was performed, as well as a motion quality check, but was the motion in the schizophrenia group significantly higher than the control group even after these procedures? And if motion is used as a nuisance regressor, is the schizophrenia group still significantly associated with lower INT?

Our apologies for not making this clear. There was no significant difference in the amount of motion between the schizophrenia group and the control group either before or after scrubbing (excessive motion was an exclusion criteria). The average value before scrubbing was used as a nuisance regressor in all analyses (although results are the same with or without the nuisance regressor). We chose to use the pre-scrubbing average motion because we believe it better represents the quality of the scan and potential impact of motion since motion scrubbing is not a perfect procedure (Power, Schlaggar, and Petersen, Neuroimage, 2015). Also, similar FD effects were observed in the HCP sample if either the pre- or post-scrubbing average FD was used as the regressor.

In the simulations, were w_EE_ in V1, V2 and V4 and also V2, V4 and MT increased by 10%, 5% and 2.5% respectively in both cases? Unless I misunderstand something there must be a misprint here, do you mean 2.5%, 5% and 10% for the latter set of areas?In any case, are the effects in Figure 4 of the same order of magnitude as the effects observed in the fMRI data? The upper and lower hierarchy effects are also quite different to each other. Why not simulate effects of similar orders of magnitude to the detected effects, this would convey what magnitude of changes to these parameters might be needed to cause these pathologies… Also, I would have thought the most realistic simulations would in fact be ones in which w_EE_ decreased throughout the hierarchy but in differing amounts depending on hierarchical level in the hallucination vs delusion cases. The simulation as it is has INT increasing above “normal” in the pathological cases, which is not what was observed in any area in the schizophrenia group, unless I'm mistaken?

We thank the reviewer for pointing this out. In the revised manuscript we now directly fit the biophysical model to the predicted changes in INT for hallucinations only, delusions only, and both hallucinations and delusions. Critically, we fit the change in INT (ΔINT) compared to the fitted values for patients with no hallucinations or delusions. We chose to take the unaltered model as the baseline for the biophysical modeling because our data and analyses are designed to investigate relative change within patients with schizophrenia, but assuming a starting point of decrease in w_EE_ does not meaningfully change any of the results or conclusions. In any case, the results of the simulation are mostly useful in terms of illustrating plausible mechanisms but the ability to draw quantitative conclusions is limited by several factors, including the fact that this model is based on non-human primate anatomical connections which lacks detail in several relevant connections such as those in the auditory system.

[Editors’ note: what follows is the authors’ response to the second round of review.]

Overall the reviewers and I were impressed by your revision. As you will see however, there was variability in how convinced the reviewers were by the new results given that they depend on a new hierarchy. In the consultation session among reviewers, one reviewer noted that while the paper is innovating and exciting, they are troubled because they felt that links between brain and behavior should be tethered/grounded at both ends. They would like to be more sure that the new way that you have chosen to define the brain hierarchy isn't the one that happens to correlate with delusions, noting " would like to be convinced that there is independent validation of the hierarchy and that is indeed the one that we find in the brain, rather than the one that happens to work best for the authors' purpose".

We thank the reviewer for raising this concern as we fully appreciate the importance of defining hierarchies in an unbiased manner. Based on the reviewers’ comments from the first round of reviews, we indeed aimed to devise an unbiased method to determine hierarchies that could be justified a priori without regard to our aims or circularity. We thought of this as an entirely independent, basic question: how can we determine the hierarchical orderings that best capture hierarchical gradients of specific sensory systems in humans, in general. First, we wanted an approach that could empirically determine the hierarchies and resolve some of the ambiguities that currently exist in the primate anatomy literature. We reviewed the primate literature to determine appropriate measures to define these hierarchies. Although several measures of hierarchy have been previously proposed (e.g. receptive-field size or tissue composition), the most widely accepted measure of hierarchy is calculated using invasive neuronal tract-tracing methods to determine the fraction of supragranular layer neurons – the number of feedforward vs. feedback projections in a given brain region (i.e. is it more feedforward or feedback). Unfortunately, these data are only available for nonhuman primates but not for humans. Despite the substantial overlap in macaque and human neuroanatomy, the lack of a one-to-one correspondence suggested the need for proxy measures for hierarchy in humans. Based on a detailed review of the literature, we determined that T1w/T2w (a ratio used to derive “myelin maps”) and cortical thickness are the two most extensively characterized and validated proxy measures of hierarchy in humans. As mentioned in the paper, the use of these measures is supported by the classic use of myeloarchitecture and cytoarchitecture in anatomical studies of cortical parcellation. But it is the extensive validation of these specific measures, which we omitted in the previous version of the manuscript, that positions them ideally to define hierarchies. In particular, Burt et al., 2018, validated T1w/T2w in macaques against the gold-standard tract-tracing measure of hierarchy, showing excellent agreement between the two (parcel-wise correlation = -0.78). Burt et al. went on to provide a comprehensive and compelling validation in macaques and humans of T1w/T2w and cortical thickness, using several measures including human postmortem gene-expression data from the Allen Human Brain Atlas (in particular granular layer-IV-specific gene expression, a proxy for cytoarchitecture structural type and the gold-standard tract-tracing measure of hierarchy), which established these two MRI measures as well justified candidates for our purpose. Note that Burt et al. already demonstrated that these measures (T1w/T2w and cortical thickness) capture a brain-wide hierarchy, but they did not use them to define hierarchies for separate sensory systems.

Having determined which proxy MRI measures to use for defining these hierarchies in humans, we determined what parcels to include in the sensory hierarchies and how to validate the specific ordering. We decided it was critical to maximize the variance of measures across the hierarchy through model selection and by including downstream prefrontal cortex regions. This second point is critical because adding prefrontal cortex regions downstream from sensory cortex to the hierarchies should a priori maximize our ability to capture hierarchical gradients in general. If present, this should also increase our ability to detect any changes in these hierarchies as a function of symptom severity by increasing the dynamic range, but without biasing results towards detecting symptom effects. Indeed, several previous studies in nonhuman primates have examined hierarchical effects of perceptual decisions and intrinsic timescales by examining these types of extended hierarchies including regions within sensory cortex and in downstream prefrontal cortex (de Lafuente and Romo Proc Natl Acad Sci 2006, Murray et al., 2014, and van Vugt et al., Science 2018 among others). This inclusion of prefrontal downstream regions was overlooked in our initial manuscript and was included in the revised version in response to an excellent suggestion by one of the reviewers, which was partly motivated by the prior literature supporting an involvement of the prefrontal cortex in the pathophysiology of delusions.

We then determined which were the most likely orderings for each hierarchy based on a review of the anatomy literature. The reason for this is that we wanted a robust data-driven validation that was informed by anatomically plausible orderings and did not consider implausible orderings at odds with the extensive neuroanatomical studies in primates (e.g. we can say with high certainty that V1 is not the highest level of the visual hierarchy and so it is not worth considering this option as plausible). We specifically came up with the 4 most likely candidate orderings for each hierarchy, which we took to reflect the main ambiguities with respect to plausible orderings of regions in human sensory hierarchies.

Having determined the appropriate MRI measures and the general method to provide an anatomically informed, data-driven refinement of the candidate hierarchies, we decided to use an independent sample (HCP data) from our schizophrenia subjects to prevent circularity. Also, the HCP dataset allowed us to use the T1w/T2w measure, which was not available in our other dataset. Note that because these structural measures are only used to define the hierarchies and not to test our hypothesis regarding the hierarchical model of psychosis, this is a completely independent test. Given all this, we firmly believe that our method represents the most accurate method we could have used to determine hierarchical orderings based on MRI data, regardless of our aims, and that our approach is well justified and unbiased with respect to our main hypothesis. And unlike much work investigating the hierarchical organization of the brain, that assumes a particular hierarchy we are actually testing and validating the hierarchical orderings.

We indeed provide evidence that the selected hierarchies based on our method capture functional hierarchical gradients in two independent datasets (resting-state-fMRI-based INT from the HCP dataset and, separately, from the controls in the combined schizophrenia dataset), at least in the auditory and somatosensory systems.

To further address this concern, we now provide an additional validation of the selected hierarchies using human postmortem gene-expression data from the Allen Human Brain Atlas. Based on Burt et al., we focused on the average expression of genes preferentially expressed in granular layer IV, a marker of cytoarchitecture structural type that indicates how granular or agranular a region is and which is to our knowledge the closest proxy for the goldstandard tract-tracing measure of hierarchy in humans. Similar to the brain-wide validation provided by Burt et al., we found that genes preferentially expressed in granular layer IV were predictive of the selected hierarchical orderings in all three sensory systems (see Figure 1—figure supplement 3). Again, we thought it was worth confirming this in our hierarchies only because Burt et al. showed the measures to correlate across the whole brain but not separately by sensory system. Therefore, this provides a third independent validation that our orderings capture substantial variance associated with biological gradients reflective of hierarchies. We thus consider this a comprehensive validation of our method that shows that, at the very least, our selection approach provided a reasonable definition to capture variance associated with hierarchical gradients.

Finally, to further convince ourselves of the hallucination and delusion effects on hierarchical gradients, particularly in the auditory system, we also used an anatomically agnostic definition of the auditory hierarchy – voxels within the auditory hierarchy parcels were binned according to their INT values in the HCP subjects. This approach also produced significant hierarchical-gradient effects of both hallucinations and delusions, with the effect for delusions even numerically greater than that of hallucinations. These results, reported in the paper (see Figure 3—figure supplement 5), also indicate that our main findings are robust to the definition of the hierarchies, which we further support below.

The following was added to the revised manuscript to address this comment:

Results:

“In particular, Burt et al., 2018, validated T1w/T2w in macaques by showing strong agreement with a goldstandard tract-tracing measure of hierarchy. […] Consistent with this, expression of granular layer IV genes showed strong, negative correlations with hierarchical level in all three winning hierarchies (auditory: r_s_ = 0.88, P = 0.003; visual: r_s_ = -0.75, P = 0.026; somatosensory: r_s_ = 0.87, P = 0.005; Figure 1—figure supplement 3).”

They then followed this up noting that your winning visual hierarchy falls within the null distribution of model fits for myelin/thickness (Figure 1B). So it seems visual hierarchy is not very reliably measured at all. But the intrinsic timescale results weren't significant in the visual system either so that doesn't really matter. They noted "If anything I think they should stop treating all the sensory hierarchies similarly and point out the visual one seems quite different but this is a side issue".

We appreciate the reviewers noticing this and agree that it is a very interesting and surprising result in and of itself. We have checked that our T1w/T2w maps and cortical thickness maps are similar to others in the literature and have found very strong relationships with others, suggesting that this phenomena of a limited hierarchical gradient of myelin content in the visual cortex has been observed several times (if only indirectly) but not thoroughly investigated. We have revised the manuscript to more clearly point out our uncertainty with regard to the selected visual system hierarchy based on the structural MRI measures used here (or by INT for that matter). Interestingly, while the visual system hierarchy including prefrontal cortex regions was predicted by layer-IV gene expression, this was not the case in the visual cortex (without prefrontal regions). This is an interesting finding that should be studied further; especially given the canonical nature of the visual hierarchy in the literature. Importantly, we do not see this as a problem with our hierarchy selection method but rather as an indication that hierarchical measures (structural, genetic, and functional) do not show very pronounced gradients within visual cortex. Even if we focus on the earliest levels of the hierarchy regardless of our selected orderings, which are uncontroversially arranged from V1 to V2 and V3, we do not see a clear alignment of the visual hierarchy with the gene expression data. For INT, we also see that V4 has lower values than V3 in two datasets, which goes against the predicted hierarchy. Similarly, for T1w/T2w, V6 has higher myelin content than V3 contrary to predictions. These examples illustrate that visual cortex does not demonstrate hierarchical gradients that can be reliably captured with these measures, at least compared to other systems. We now acknowledge this limitation more clearly. But since we suspected from the previous version of the manuscript that effects would be stronger for the non-visual systems (perhaps because we can more reliably define the hierarchies in these systems) we thought removing the visual system for the main hypothesis tests would inflate our results, so we decided to keep the visual hierarchy along with the other systems to provide a more stringent and unbiased test. Because each sensory system is included as a separate regressor and we test for interactions between them in our main model (M2), we make no assumptions that the systems are similar and treat them independently.

The following was added to the revised manuscript to address this comment:

Results:

“Positive but non-significant correlations were observed in the visual system (in-sample: r_s_ = 0.27, P =0.49, P_permutation_ = 0.47; out-of-sample: r_s_ = 0.22, P = 0.58; Figure 1C). […] Thus, we empirically validated extended sensory hierarchies that captured variability in hierarchical indices across three independent datasets, although this was generally less clear for the visual system.”

Discussion:

“Our null findings in the visual system are also qualified by the poorer correspondence between levels of the visual hierarchy and hierarchical MRI indices (not only for INT but also surprisingly for the structural indices) compared to the other systems (Figure 1). This suggests the need for further investigation into the sensitivity of available MRI measures of hierarchy to uncover the underlying gradients within the visual cortex.”

Given these divergent opinions but with overall positive inclinations, I would like you to consider some more moderate way you could address this, e.g. by reporting more fully the AVH and delusion affects for all 4 auditory rankings and discussing the implications of the revised approach which then leads to relation to delusions.

We appreciate the opportunity to fully address this concern. We have added a figure supplement for Figure 3 that shows the t-statistics for the hierarchical-gradient effects of hallucination and delusion severities, and their interactions for each of the 4 auditory, visual and somatosensory orderings we considered based on the anatomy literature. As can be seen, the results are similar for the selected and 3 non-selected orderings, indicating the results were robust to our selection approach. Indeed, a family-wise error-corrected test including all 4 orderings by system found that our observation of significant negative hierarchical-gradient effects of hallucinations in 4 out of 12 orderings, significant positive hierarchical-gradient effects of delusions in 6 out of 12 orderings, and significant positive interactions between hierarchical-gradient effects of hallucinations and delusions in 8 out of 12 orderings was statistically above chance (set-level *P_permutation_* = 0.009).

As for the implications of the revised hierarchy-selection approach, we feel that this approach is better because it is more principled and less circular. We are confident that this approach does not induce a bias towards us finding the hypothesized results. We believe that our approach, and particularly the inclusion of downstream prefrontal regions, increased our sensitivity to detect (true) effects by maximizing the variance related to hierarchical gradients in our main measure of interest, namely INT (even if INT data was not used for selecting the orderings). As mentioned above, inclusion of prefrontal regions was suggested by a reviewer and is common in hierarchical studies to capture full hierarchies (from earliest to highest levels of the hierarchy). In essence, we believe this approach increases the dynamic range in our functional hierarchy measure, which therefore increased our statistical power. In addition, including prefrontal regions can be justified by the previous literature involving prefrontal cortex in the pathophysiology of delusions. But it is notable that our main findings on INT clearly represent continuous changes in the slope of hierarchical gradients rather than effects driven only by the lowest or highest level of the hierarchy (even if the effects can be modeled by localized changes in E/I ratio), suggesting that inclusion of prefrontal regions did not induce an artifact but afforded increased power to detect gradient changes.

The following was added to the revised manuscript to address this comment:

Results:

“To rule out an effect of our approach for selecting hierarchical orderings on these results, we tested these symptom effects for each of the 4 different sensory cortex hierarchical orderings considered a priori candidates for each sensory system. Results were generally consistent across the different hierarchical orderings (Figure 3—figure supplement 2), particularly in the auditory system. A family-wise permutation test similar to the one above, but including all 4 orderings per system (12 total orderings), showed that the observed set of results was statistically above chance for all systems (set-level P_permutation_ = 0.002) and for the auditory system alone (set-level P_permutation_ = 0.001).”

Reviewer #2:This revision and appeal is much improved.It is challenging since we should not moderate our enthusiasm for a piece based on the specific results, however, the fact that the gradients now relate significantly and oppositely to hallucinations and delusions is encouraging.Here is my remaining concern. The authors can't have it both ways. They reclassified the hierarchy and got this interesting and compelling pattern of findings. The pattern is even significant compared to a random ordering of regions. However, I would like to be reassured further that:1) This is the most appropriate construction of hierarchy, i.e.the choice of hierarchy construction reflects biological reality (leveraging for example postmortem data on which there are also MRI data).

We thank the reviewer for raising this important concern. Please see our main response above.

2) What impact the choice of hierarchy construction has on the symptom associations, that is, compared to some control other than random, how robust are the associations, given that they made some different choices and got a less robust set of effects.

We appreciate the reviewer raising this concern. We agree that the choice of hierarchy construction has a relevant role in determining the symptom effects that we hypothesized and observed. Here, however, we made an effort to use the most principled and unbiased method to define the hierarchies, as indicated above. We believe that the change that made the biggest difference was adding the prefrontal regions (in response to a comment from this reviewer), which as we mention above, increased the dynamic range of the hierarchy and our statistical power to detect symptom-related changes in hierarchies. We also presented data above showing that results using an anatomically agnostic hierarchical ordering within auditory cortex were consistent with our main results, suggesting their robustness.

To further support the robustness of our results to other reasonable choices of hierarchical orderings, we have conducted an additional analysis looking at the 4 orderings we used for each sensory system based on the anatomy literature. These results (below) show that the main results are robust to this choice. They are now included as a figure supplement for Figure 3 that shows similar t-statistics for the hierarchical-gradient effects of hallucination and delusion severities, and their interactions for each of the 4 auditory, visual and somatosensory orderings. Accounting for all orderings and systems, a family-wise error-corrected test including all 4 orderings by system found that our observation of significant negative hierarchical-gradient effects of hallucinations in 4 out of 12 orderings, significant positive hierarchical-gradient effects of delusions in 6 out of 12 orderings, and significant positive interactions between hierarchical-gradient effects of hallucinations and delusions in 8 out of 12 orderings was statistically above chance (set-level *P_permutation_* = 0.009).

The following was added to the revised manuscript to address this comment:

“Results:

To rule out an effect of our approach for selecting hierarchical orderings on these results, we tested these symptom effects for each of the 4 different sensory cortex hierarchical orderings considered a priori candidates for each sensory system. Results were generally consistent across the different hierarchical orderings (Figure 3—figure supplement 2), particularly in the auditory system. A family-wise permutation test similar to the one above, but including all 4 orderings per system (12 total orderings), showed that the observed set of results was statistically above chance for all systems (set-level P_permutation_ = 0.002) and for the auditory system alone (set-level P_permutation_ = 0.001).”

Reviewer #3:I think the authors have done a great job in responding to the comments and the paper is definitely stronger as a result. I have only a couple of comments.I have some trouble understanding the new modelling part, the description is not clear in the text and neither in the figure legend. The different panels in Figure 4 are also not explicitly referenced in the text (at least not in the rebuttal letter). There is also not much labelling in Figure 4 itself. Could this all please be clarified?

We apologize for the lack of clarity and thank the reviewer for giving us the opportunity to revise for clarification. We have also modified Figure 4 for improved clarity.

“Results:

Altered E/I Ratio as a Potential Biological Mechanism

To explore candidate biological mechanisms for the effects we observed in vivo, we leveraged a largescale biophysical model previously shown to capture intrinsic timescale hierarchies (Chaudhuri et al., 2015). […] This suggests that additivity of the local symptom-specific alterations could explain symptom co-occurrence.”

Materials and methods:

“The E/I ratio changes were modeled as a triangle function where a local maximum exhibited a peak E/I ratio increase and other nodes had E/I ratio changes that decreased linearly as a function of absolute distance in hierarchical levels from the peak. […] This was done by calculating the error between the biophysical model with E/I ratio changes for the hallucination parameters and exemplary case 2; the error between the biophysical model with E/I ratio changes for the delusion parameters and exemplary case 3; the error between the biophysical model with E/I ratio changes determined by the sum of the E/I ratio changes for the hallucination parameters and the E/I ratio changes for the delusion parameters, and exemplary case 4; and minimizing the sum of squared errors.”

Some specific issues too:The authors state "the best-fitting levels of the peak increase in local E/I ratio were levels 1 and 8" but this is a six node hierarchy? Should this be levels 1 and 6?

We appreciate the reviewer allowing us to clarify this issue. While we only use 6 nodes for the biophysical-model hierarchy, we assigned the hierarchical levels of the nodes according to their corresponding hierarchical levels in vivo to capture the underlying spacing between levels of the hierarchy even if we need to skip some levels. Because retrograde tracer data are not yet available for all brain regions, our hierarchy is missing levels 3, 6, and 7 (regions V3, V6, and V7 respectively). But we still wanted our hierarchy scale to reflect, for instance, that level 8 is more separate from level 5 than level 2 is from level 1. We have further clarified this in the revised manuscript.

The following was added to the revised manuscript to address this comment:

“Results:

We specifically used the 6 biophysical-model nodes that directly corresponded to levels of our visual hierarchy and for which tract-tracing data were available: V1 (level 1), V2 (level 2), V4 (level 4), MT (level 5), 8l (level 8), and 46d (level 9).”

The in silico plots in Figure 4B look identical all along the row. Is that meant to be the case?

We thank the reviewer for pointing this out. While the in silico plots do look very similar along the row, there are subtle differences throughout; the differences are difficult to display because of the large range of values between the lowest and highest nodes. Given the difficulty in visualizing this difference and to simplify the figure, we have removed Figure 4B.

I'm also not clear why in vivo auditory results are being compared with in silico visual ones?

We thank the reviewer for raising this issue. In vivo auditory results were used because this was the hierarchy where we observed the clearest effects for hallucinations and delusions. In silico visual results were used because retrograde tracer data are not available for the majority of auditory cortex regions. Similarly, retrograde tracer data are not available for the lowest levels of the somatosensory hierarchy (areas 3b and 3a) hindering our ability to sufficiently model the entire hierarchy. While we agree that this is not ideal, the dynamics of the biophysical model should be similar along the hierarchies for each of the sensory systems. We would further like to clarify that the results of the model fitting (i.e., the magnitude of E/I ratio changes) are most helpful to provide qualitative insights but cannot be taken to reflect the biological system in a quantitatively accurate way. This is because, even if auditory nodes were available, the biophysical model uses anatomical macaque data rather than human data. On the contrary, the qualitative patterns and general conclusions (i.e., that there are local E/I ratio changes and a hierarchical gradient of effects) should hold true regardless of which sensory hierarchy in macaques is used and if a human model is used. In other words, we are using the macaque visual hierarchy as a model hierarchy with biophysical and anatomically realistic constraints to reproduce observed effects and gain qualitative insights on candidate underlying processes.

The following was added to the revised manuscript to address this comment:

“Results:

We modeled the in vivo DINT in the auditory system using the macaque visual system as a model hierarchy with realistic biological constraints due to the lack of tract-tracing data for the auditory system; note that sensory system and species differences limit our ability to derive precise quantitative conclusions from the modeling results but still afford qualitative insights. We specifically used the 6 biophysical-model nodes that directly corresponded to levels of our visual hierarchy and for which tracttracing data were available: V1 (level 1), V2 (level 2), V4 (level 4), MT (level 5), 8l (level 8), and 46d (level 9).”

The legend descriptions "insets for A" and "Insets for B" should be B and C respectively, I think?

We thank the reviewer for bringing this to our attention. The revised legend has been corrected.

Also in the phrase "Insets for A show predicted INT values" does “predicted” mean estimated from in vivo data? “Predicted” sounds like a model has been involved but I assume that is not the case?

We appreciate the reviewer raising this question and apologize for the lack of clarity. The in vivo data displayed are from “exemplary cases” based on regression fits of model M1_primary_. From this model we can estimate INT values per hierarchical level from exemplary cases capturing the extreme symptom profiles (high hallucination only, high delusion only, both high, or both low), which allows us to produce more idealized and extreme patterns that facilitate model fitting. We have revised the manuscript to improve the clarity of this section.

The following was added to the revised manuscript to address this comment:

“Results:

To fit the biophysical model, we first estimated in vivo data for exemplary cases using regression fits from M1_primary_ (Materials and methods) in the auditory system – the system that showed the strongest effects. […] For all exemplary cases, the severity of other symptoms and the values of covariates were set to the average values from all patients. Changes of INT for exemplary cases 2–4 were determined as the difference in INT relative to the ‘no hallucinations or delusions’ case (in vivo DINT; Figure 4A).”

I don't understand the difference between the Insets for A and B?

We apologize again for the lack of clarity. We have modified this figure to no longer include the insets. Figure 4A now shows the difference in estimated INT values (from regression fits of model M1_primary_) between the 3 pathological exemplary cases (hallucinations, delusions, or both) and the reference exemplary cases (no hallucinations or delusions). Figure 4B now shows the biophysical model. Figure 4C now shows the difference in simulated INT values (from the biophysical model) between the three pathological scenarios and the reference model. Figure 4D now shows the changes in E/I ratio used to produce the results in Figure 4C (determined by fitting the results in Figure 4A).